

**A physically-based distributed karst hydrological model (QMG**
**model-V1.0) for flood simulations**
**Ji Li[1,*], Daoxian Yuan[1,2], Fuxi Zhang[3], Yongjun Jiang[1], Jiao Liu[4], Mingguo Ma[1]**
[1]Chongqing Jinfo Mountain Karst Ecosystem National Observation and Research
Station,Chongqing Key Laboratory of Karst Environment, School of Geographical
Sciences, Southwest University, Chongqing 400715, China
[2]Karst Dynamic Laboratory, Ministry of Land and Resources, Guilin 541004, China
[3]College of Engineering Science and Technology, Shanghai Ocean University;
Shanghai Engineering Research Center of Marine Renewable Energy 201306, China
[4]Chongqing municipal hydrological monitoring station, Chongqing 401120, China
Corresponding author: Ji Li (445776649@qq.com)
**Abstract** Karst trough valleys are prone to flooding, primarily because of the unique
hydrogeological features of karst landform, which are conducive to the spread of rapid
runoff. Hydrological models that represent the complicated hydrological processes in karst
regions are effective for predicting karst flooding, but their application has been hampered
by their complex model structures and associated parameter set, especially so for distributed
hydrological models, which require large amounts of hydrogeological data. Distributed
hydrological models for predicting the Karst flooding is highly dependent on distributed
structrues modeling, complicated boundary parameters setting, and tremendous
hydrogeological data processing that is both time and computational power consuming.
Proposed here is a distributed physically-based karst hydrological model, known as the
QMG (Qingmuguan) model. The structural design of this model is relatively simple, and it is





generally divided into surface and underground double-layered structures. The parameters
that represent the structural functions of each layer have clear physical meanings, and the
parameters are less than those of the current distributed models. This allows modeling in
karst areas with only a small amount of necessary hydrogeological data. 18 flood processes
across the karst underground river in the Qingmuguan karst trough valley are simulated by
the QMG model, and the simulated values agree well with observations, for which the
average value of Nash–Sutcliffe coefficient was 0.92. A sensitivity analysis shows that the
infiltration coefficient, permeability coefficient, and rock porosity are the parameters that
require the most attention in model calibration and optimization. The improved
predictability of karst flooding by the proposed QMG model promotes a better mechanistic
depicting of runoff generation and confluence in karst trough valleys.
**Keywords:** Simulation and forecasting of karst floods; Karst trough valleys; QMG
(Qingmuguan) model; Parametric optimization; Parameter sensitivity analysis
**1. Introduction**

Karst trough valleys are very common in China, especially in the southwest. In general,

these karst areas are water scarce because their surfaces store very little rainfall, but it is also
a protential birthplace for floods. Because trough and valley landforms and topographic
features facilitate the formation and propagation of floods (Li et al., 2021). Taking the study
area, the Qingmuguan karst trough valley for example, floods used to happen here
constantly. In recent years, with the increase in extreme rainfall events and the increased
area of construction land in the region, rainfall infiltration decreased as long as rapid runoff
over impervious surfaces increased that results in frequent catastrophic flooding in the basin
(Liu et al., 2009). Excess water overflows from karst sinkholes and underground river
outlets often occur during floods, flooding large areas of farmland and residential areas and
causing serious economic losses (Yu et al., 2020). Therefore, the simulation and prediction





of karst flooding events in these karst trough valleys like the study area are both important
and urgently needed.
Hydrological models can be effective for forecasting floods and evaluating water
resources in karst areas (Ford and Williams, 2007; Williams, 2009). However, modeling
floods in karst regions is extremely difficult because of the complex hydrogeological
structure. Karst water-bearing systems consist of multiple media under the influence of
complex karst development dynamics (Worthington et al., 2000; Kovács and Perrochet,
2008), such as karst caves, conduits, fissures, and pores, and are usually highly spatially
heterogeneous (Chang and Liu, 2015; Mario et al., 2019). In addition, the intricate surface
hydrogeological conditions and the hydrodynamic conditions inside the karst water-bearing
medium result in significant temporal and spatial differences in the hydrological processes in
karst areas (Geyer et al., 2008; Bittner et al., 2020).
Early studies on flood forecasting in karst regions, with simplified lumped hydrological
models , were commonly used to describe the rainfall–discharge relationship (e.g., Kovács
and Sauter, 2007; Fleury et al., 2007b; Jukić and Denić, 2009; Hartmann et al., 2014a). With
the development of physical exploration technology and the progress made in mathematics,
computing, and other interdisciplinary disciplines, the level of modeling has gradually
improved (Hartmann and Baker, 2017; Hartmann, 2018; Petrie et al., 2021). Distributed
hydrological models have subsequently become widely used in karst areas. The main
difference between the lumped and distributed hydrological models is that the latter divide
the entire basin into many sub-basins to calculate the runoff generation and confluence,
thereby better describing the physical properties of the hydrological processes inside the
karst water-bearing system (Hartmann, 2018; Epting et al., 2018).
Because of their simple structure and little demand for modeling data, lumped
hydrological models have been used widely in karst areas (Kurtulus and Razack, 2007;
Ladouche et al., 2014). In a lumped model, the river basin is considered as a whole to
calculate the runoff generation and confluence, and there is no division running into
sub-basins (Dewandel et al., 2003; Bittner et al., 2020). Lumped models usually consider the
inputs and outputs of the model (Liedl and Sauter, 2003; Hartmann and Bake, 2013, 2017).



In addition, most of the model parameters are not optimized in a lumped model, and the
physical meaning of each parameter is unclear (Chen, 2009; Bittner et al., 2020).

Distributed hydrological models are of active interest in flood simulation and

forecasting research (Ambroise et al., 1996; Beven and Binley, 2006; Zhu and Li, 2014).
Compared with a lumped model, a distributed model has a more definite physical
significance for the model structure in terms of its mechanism (Meng and Wang, 2010;
Epting et al., 2018). In a distributed hydrological model, an entire karst basin can be divided
into many sub-basins (Birk et al., 2005) using high-resolution digital elevation map (DEM)
data. In the rainfall-runoff algorithm of the model, the hydrogeological conditions and karst
aquifer characteristics can be fully considered to precisely simulate the runoff generation
and confluence (Gang et al., 2019). The commonly used basin distributed hydrological
models (i.e., not a special groundwater numerical model such as MODFLOW) have also
been applied widely in karst areas, and include the SHE/MIKE SHE model (Abbott et al.,
1986a,b; Doummar et al., 2012), SWMM model (Peterson and Wicks, 2006; Blansett and
Hamlett, 2010; Blansett, 2011), TOPMODEL (Ambroise et al., 1996; Suo et al., 2007; Lu
et al., 2013; Pan, 2014), and SWAT model (Peterson and Hamlett, 1998; Ren, 2006).

The commonly used distributed hydrological models have multiple structures and

numerous parameters (Lu et al., 2013; Pan, 2014), which make distributed model may need
vast amounts of data to build its framework in karst regions. For example, the distributed
groundwater model MODFLOW-CFPM1 requires detailed data regarding the distribution of
karst conduits in a study area (Reimann et al., 2009; Qin and Jiang, 2014). Another example
is the Karst–Liuxihe model (Li et al., 2019), which has 5 underground vertical layers in the
model structure and has 15 parameters, make it hard to model in karst areas. In addition, a
special borehole pumping test may be required to obtain the rock permeability coefficient.

To overcome the difficulty of the large modeling-data demands for distributed

hydrological models in karst areas, a new physical model based on distributed hydrological
known as the QMG (Qingmuguan) model was developed. This QMG model has a
double-layer structure and fewer parameters. The horizontal structure is divided into river
channel units and slope units. The vertical structure below the surface is divided into a





shallow karst aquifer and a deep karst aquifer system. Only a small amount of
hydrogeological data is needed for modeling in karst basins.
To ensure that the QMG model work well in karst flood simulation and prediction in the
case of relatively simple structure and parameters. We carefully designed the algorithms of
runoff generation and confluence in the model. And to verify its applicability of QMG model
in flood simulation in karst basins, we selected the Qingmuguan karst trough valley in
Chongqing, China, as the study area for a flood simulation and uncertainty analysis. In
particular, the sensitivity of model parameters was analyzed.
**2. Study Area and Data**
2.1. Landform and topography
The Qingmuguan karst trough valley is located in the southeastern part of the Sichuan
Basin, China, at the junction of the Beibei and Shapingba districts in Chongqing, with
coordinates of 29°40′N ~ 29°47′N, 106°17′E ~ 106°20′E. The basin covers an area of 13.4
km$^2$ and is part of the southern extension of the anticline at Wintang Gorge in the Jinyun
Mountains, with the anticlinal axis of Qingmuguan located in a parallel valley in eastern
Sichuan (Yang et al., 2008). The surface of the anticline is heavily fragmented, and faults
are extremely well-developed with large areas of Triassic carbonate rocks exposed. Under
the long-term erosion of karst water, a typical karst landform pattern of "three mountains
and two troughs" has formed (Liu et al., 2009). Such karst trough landform provides
convenient conditions for flood propagation, and the karst landform development is
extremely common in the karst region of southwest China, especially in the karst region of
Chongqing. Similar regions include the karst trough valley of the Zhongliang Mountains and
the Laolongdong karst basin in Nanshan, Chongqing.
The basin is oriented north-north-east and south-south-west in a narrow band of slightly
curved arcs and is about 12 km long from north to south. The direction of the mountains in
the region is basically the same as the direction of the tectonic line. The difference in
relative elevation is between 200–300 m. Fig.1 is a map showing an overview of the

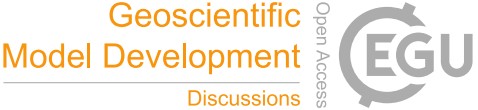

Qingmuguan karst basin.

Figure 1 The Qingmuguan karst basin.

2.2. Hydrogeological conditions

The Qingmuguan basin is located within the subtropical humid monsoon climate zone,

with an average temperature of 16.5℃ and a average precipitation of 1250 mm that is
mainly concentrated in May-September. An underground river system has developed in the
karst trough valley, with a length of 7.4 km, the water supply of underground river mainly
comes from rainfall recharge (Zhang, 2012). Most of the precipitation is collected along the
hill slope into the karst depressions at the bottom of the trough valley, where it is recharged
to the underground river through the dispersed infiltration of surface karst fissures and
concentrated injection from sinkholes (Fig. 1a). An upstream surface river collects in a
gentle valley and enters the underground river through the Yankou sinkhole (elevation 524
m). Surface water in the middle and lower reaches of the river system mainly enters the
underground river system through catenuliform sinkholes or fissures.

The stratigraphic and lithologic characteristics of the basin are largely dominated by

carbonate rocks of the Lower Triassic Jialingjiang Group ($T_{1j}$) and Middle Triassic Leikou
Slope Group ($T_{2l}$) on both sides of the slope, with some quartz sandstone and mudstone
outcrops of the Upper Triassic Xujiahe Group ($T_{3xj}$) (Zhang, 2012). The topography of the
basin presents a general anticline (Fig. 1b), where carbonate rocks on the surface are
corroded and fragmented, with a large permeability coefficient. Compared with the core of
the anticline, the rocks of the two wings of the anticline are less eroded and form a good
waterproof layer.

To investigate the karst conduits distribution in the underground river system, we

conducted a tracer test in the study area. The tracer was placed into the Yankou sinkhole and
recovered in the Jiangjia spring (Fig. 1a,c). According to the tracer test results (Gou et al.,
2010), the karst water-bearing medium in the aquifer was anisotropic, whereas the soluble
carbonate rocks were extremely permeable. The karst conduits in the underground river





were extremely well-developed, and there was a large single-channel underground river. The
response time of the underground river to rainfall was very fast, with the peak flow observed
at the outlet of Jiangjia spring 6–8 hours after rainfall. The flood peak rose quickly and the
duration of the peak flow was short. The underground river system in the study area is
dominated by large karst conduits, which is not conducive to water storage in water-bearing
media, but is very conducive to the propagation of floods.
2.3. Data

To build the QMG model to simulate the karst flood events, the necessary modelling

baseline data had to be collected, including: 1) high-resolution DEM data and
hydrogeological data (e.g., the thickness of the epikarst zone, rainfall infiltration coefficient
on different karst landforms, and permeability coefficient of rock); 2) land-use and soil type
data; and 3) rainfall data in the basin and water flow data of the underground river. The
DEM data was downloaded from a free database on the public Internet, with an initial spatial
resolution of $30 \times 30$ m. The spatial resolution of landuse and soil types were $1000 \times 1000$
m, and they were also downloaded from the Internet. After considering the applicability of
modelling and computational strength, as well as the size of the basin in the study area (13.4
$km^2$), the spatial resolution of the three types of data was resampled uniformly in the QMG
model and downscaled to $15 \times 15$ m based on a spatial discrete method by Berry et. (2010).

The hydrogeological data necessary for modelling was obtained in three simple ways. 1)

A basin survey was conducted to obtain the thickness of the epikarst zone, which was
achieved by observing the rock formations on hillsides following cutting for road
construction. Information was collected regarding the location, general shape, and size of
karst depressions and sinkholes, which had a significant impact on compiling the DEM data
and determining the convergence process of surface runoff. 2) Empirical equations
developed for similar basins were used to obtain the rainfall infiltration coefficient for
different karst landforms and the permeability coefficient of rock. For example, the rock
permeability coefficient was calculated based on an empirical equation from a pumping test
in a coal mine in the study area (Li et al., 2019). 3) A tracer experiment was conducted in the





study area (Gou et al., 2010) to obtain information on the underground river direction and
flow velocity.
Rainfall and flood data are important model inputs, and represent the driving factors
that allow hydrological models to operate. In the study area, rainfall data was acquired by
two rain gauges located in the basin (Fig. 1a). Point rainfall was then spatially interpolated
into basin-level rainfall (for such a small basin area rainfall results obtained from two rain
gauges was considered representative). There were 18 karst flood events in the period of 14
April 2017 to 10 June 2019. We built a rectangular open channel at the underground river
outlet and set up a river gauge on it (Fig. 1a) to record the water level and flow data every 15
minutes.

**3. Methodology**

**3.1. Hydrological model**

The hydrological model developed in this study was named the QMG model after the
basin for which it was developed and to which it was first applied, i.e., the Qingmuguan
basin. The QMG model proposed in this study has a two-layer structure, including a surface
part and an underground part, with the former mainly performing the calculation of runoff
generation and the confluence of the surface river, while the latter performs the confluence
calculation of the underground river system.
The structure of the QMG model is divided into a two-layer structure, both horizontally
and vertically. The horizontal structure of the model is divided into river channel units and
slope units. The vertical structure below the surface is divided into a shallow karst aquifer
(including soil layers, karst fissures, and conduit systems in the epikarst zone) and a deep
karst aquifer system (rock stratum and underground river system). This relatively simple
model structure makes it need only a small amount of hydrogeological data when modeling
in karst regions. Fig. 2 shows a flowchart of the modeling and calculation procedures
required for the QMG model.
Figure 2 Modeling flow chart of QMG (Qingmuguan) model.
To accurately describe the runoff generation and confluence on a grid scale, these karst





sub-basins are further divided into a lot of karst hydrological response units (KHRUs). The
specific steps involved in the division were adopted by referring to studies of hydrological
response units (HRUs) in TOPMODEL by Pan (2014). As the smallest basin computing
units, the KHRUs can effectively ignore the spatial differences of karst development within
the units and reduce the uncertainty in the classification of model units. Fig. 3 shows the
spatial structure of the KHRUs.

Figure 3 Spatial structure of the KHRUs (Li et al.,2021).

The modeling and operation of the QMG model consists of three main stages: 1) spatial

interpolation, and the retention of rainfall and evaporation calculations; 2) runoff generation
and confluence calculation for the surface river; and 3) confluence calculation for the
underground runoff, including the confluence in the shallow karst aquifer and the
underground river system.
**3.1.1. Rainfall and evaporation Calculation**

In the QMG model, the spatial interpolation of rainfall is accomplished by a kriging

method using the ArcGIS 10.2 software. The Tyson polygon method may be a simpler
method for rainfall interpolation if the number of rainfall gauges in the basin is sufficient.
The point rainfall observed by the two rainfall gauges in the basin (Figure 1a) was spatially
interpolated into areal rainfall for the entire basin.

Basin evapotranspiration in the KHRUs was mainly vegetal, soil evaporation, and water

surface evaporation. They were calculated using the following equations (modified from Li
et al., 2020):

$$
\begin{cases}
E_v = V^{t+\Delta t} - V^t - P_v \\
E_s = \lambda E_p, \text{if } F = F_c \\
E_s = \lambda E_p \dfrac{F}{F_c}, \text{if } F < F_{sat} \\
E_w = \Delta e \cdot \left[ 1.12 + 0.62 \left( \Delta T \right)^{0.9} \right] \cdot \left[ 0.084 + 0.24 \left( 1 - \gamma^2 \right)^{1/2} \right] \cdot \left[ 0.348 + 0.5 \omega^{1.8\text{-}1.137\omega^{0.05}} \right]
\end{cases}
\tag{1}
$$

Here, $E_v$ [mm] is the vegetal discharge, $V^{t+\Delta t} - V^t$ [mm] is the rainfall variation

by vegetation interception, $P_v$ [mm] is the vegetation interception of rainfall, and $E_s$



[mm] is the actual soil evaporation. The term $\lambda$ is the evaporation coefficient. The term
$E_p$ [mm] is the evaporation capability, which can be measured experimentally or
estimated by the water surface evaporation equation $E_w$. The term $F$ [mm] is the actual
soil moisture, $F_{sat}$ [mm] is saturation moisture content, $F_c$ [mm] is field capacity, $E_w$
[mm/d] is evaporation of water surface, and $\Delta e = e_0 - e_{150}$ [hPa] is the draught head
between the saturation vapor pressure of the water surface and the air vapor pressure
150 m above the water surface (150 m above the water surface was selected here
because the altitude for temperature and humidity observations in the southwestern
karst regions of China is usually set at 150–200 m). The term $\Delta T = t_0 - T_{150}$ [°C] is the
temperature difference between the water surface and the temperature 150 m above the
water surface, $\gamma$ is the relative humidity 150 m above the water surface, and $\omega$ [m/s]
is the wind speed 150 m above the water surface.
**3.1.2. Runoff Generation**
In the QMG model, the surface runoff generation in river channel units means the
rainfall in the river system after deducting evaporation losses. This portion of the runoff will
participate in the confluence process directly through the river system, rather than
undergoing infiltration. In contrast, the process of runoff generation in slope units is more
complex, and its classification is related to the developmental characteristics of surface karst
in the basin, rainfall intensity, and soil moisture. For example, when the soil moisture
content is already saturated, there is the potential for excess infiltration surface runoff in
exposed karst slope units. The surface runoff generation of the KHRUs in the river channel
units and slope units can be described by the following equations (modified from Chen,
2009, 2018; Li et al., 2020):
$$\begin{cases} P_r(t) = \left[ P_i(t) - E_p \right] \dfrac{L \cdot W_{max}}{A} \\ R_{si} = (P_i - f_i), P_i \geq f_{max} \\ R_{si} = 0, P_i < f_{max} \\ f_{max} = \alpha (F_c - F)^\beta + F_s \end{cases} \tag{2}$$

Here, $P_r(t)$ [mm] is the net rainfall (deducting evaporation losses) in the river channel
units at time $t$, hours; $P_i(t)$ [mm] is the rainfall in the river channel units, $L$ [m] is the length





of the river channel, $W_{max}$ [m] is the maximum width of the river channel selected, and $A$
[m²] is the cross-sectional area of the river channel. $R_{si}$ is termed excess infiltration runoff in
the QMG model, [mm]; when the vadose zone is short of water and has not been filled. The
infiltration capacity $f_{max}$ is different in different karst landform units, $\alpha, \beta$ are Holtan model's
parameters; and $F_s$ [mm] is the stable depth of soil water infiltration.
In the KHRUs (Fig. 3), underground runoff is generated primarily from the infiltration
of rainwater and direct confluence recharge from sinkholes or skylights. In the QMG model,
the underground runoff is calculated by the following equations (modified from Chen,

2018):


$$\begin{cases} R_g = R_0 \exp(-pt^m) \\ R_e = v_e \cdot I_w \cdot z \end{cases} \quad (3)$$

where

$$\begin{cases} \dfrac{\partial R_e}{\partial x} + I_w \cdot z \cdot \dfrac{\partial F}{\partial t} = R_r - R_{epi} \\ v_e = K \cdot \tan(\alpha), \quad F > F_c \\ v_e = 0, \quad F \le F_c \end{cases} \quad (4)$$

Here, $R_g$ [mm] is the underground runoff depth (this part of the underground runoff is
mainly from the direct confluence supply of the karst sinkholes or skylights in the study
area), $R_0$ [mm] is the average depth of the underground runoff, $p$ and $m$ are attenuation
coefficients calculated by conducting a tracer test in the study area, $R_e$ [L/s] is the
underground runoff generated from rainfall infiltration in the epikarst zone, $I_w$ [mm] is the
width of the underground runoff on the KHRUs, $z$ [mm] is the thickness of the epikarst zone,
$R_r$ [mm²/s] is the runoff recharge on the KHRUs during period $t$, $R_{epi}$ [mm²/s] is the water
infiltration from rainfall, $v_e$ [mm/s] is the flow velocity of the underground runoff, $K$
[mm/s] is the current permeability coefficient, and $\alpha$ is the hydraulic gradient of the
underground runoff. If the current soil moisture is less than the field capacity, i.e., $F \le F_c$,
then the vadose zone is not yet full, there will be no underground runoff generation, and





rainfall infiltration at this time will continue to compensate for the lack of water in the
vadose zone until it is full and before runoff is generated.
**3.1.3. Channel Routing and Confluence**

In the QMG model, the calculation of runoff confluence on the KHRUs includes the

confluence of surface river channel and underground runoff. There are already many mature
and classical algorithms available for calculating the runoff confluence in river channel units
and slope units, such as the Saint-Venant equations and Muskingum convergence model. In
this study, the Saint-Venant equations were adopted to describe the confluence in the surface
river and hill slope units, for which a wave movement equation was adopted to calculate
confluence in slope units (Chen, 2009):
$$
\begin{cases}
\dfrac{\partial Q}{\partial x} + L\dfrac{\partial h}{\partial t} = q \\
S_f - S_0 = 0
\end{cases}
\tag{5}
$$

where
$$
Q = vhL = \frac{L}{n} h^{\frac{5}{3}} S_0^{\frac{1}{2}}.
\tag{6}
$$

Here, we customized two variables $a$ and $b$:

$$
\begin{cases}
a = (\dfrac{n}{L} S_0^{-\frac{1}{2}})^{\frac{3}{5}} \\
b = \dfrac{3}{5}
\end{cases}
\tag{7}
$$

Equation (7) was substituted into equation (5) and discretized by a finite difference

method, giving
$$
\begin{cases}
\dfrac{\partial Q}{\partial x} + abQ^{(b-1)} \dfrac{\partial Q}{\partial t} - q = 0 \\
\dfrac{\Delta t}{\Delta x} Q_{i+1}^{t+1} + a(Q_{i+1}^{t+1})^b = \dfrac{\Delta t}{\Delta x} Q_i^{t+1} + a(Q_{i+1}^t)^b + q_{i+1}^{t+1}\Delta t
\end{cases}
\tag{8}
$$

The Newton–Raphson method was used for the iterative calculation using equation (8):





$$\left[Q_{i+1}^{t+1}\right]^{k+1} = \left[Q_{i+1}^{t+1}\right]^{k} - \frac{\frac{\Delta t}{\Delta x}\left[Q_{i+1}^{t+1}\right]^{k} + a(\left[Q_{i+1}^{t+1}\right]^{k})^{b} - \frac{\Delta t}{\Delta x}Q_{i}^{t+1} - a(Q_{i+1}^{t})^{b} - q_{i+1}^{t+1}\Delta t}{\frac{\Delta t}{\Delta x} + ab(\left[Q_{i+1}^{t+1}\right]^{k})^{b-1}}, \quad (9)$$

where $Q$ [L/s] is the confluence of water flow in slope units, $L$ [dm] is its runoff width,
$h$ [dm] is the runoff depth, and $q$ [dm$^2$/s] is the lateral inflow on the KHRUs. Here, the
friction slope $S_f$ equals the hill slope $S_0$, and the inertia term and the pressure term in the
motion equation of the Saint-Venant equations were ignored. The term $v$ [dm/s] is the flow
velocity of surface runoff in the slope units as calculated by the Manning equation, $n$ is the
roughness coefficient of the slope units, $Q_i^{t+1}$ [L/s] is the slope inflow in the KHRU at time
$t+1$, and $Q_{i+1}^{t+1}$ [L/s] is the slope discharge in the upper adjacent KHRU at time $t+1$.
Similarly, the surface river channel confluence was described based on Saint-Venant
equation, where a diffusion wave movement equation was adopted, means the inertia term in
the motion equation was ignored:
$$\begin{cases} \dfrac{\partial Q}{\partial x} + \dfrac{\partial A}{\partial t} = q \\ S_f = S_0 - \dfrac{\partial h}{\partial x} \end{cases} \quad (10)$$

A finite difference method and the Newton–Raphson method were used for the iterative
calculation of the above equation:
$$\begin{cases} \left[Q_{i+1}^{t+1}\right]^{k+1} = \left[Q_{i+1}^{t+1}\right]^{k} - \dfrac{\frac{\Delta t}{\Delta x}\left[Q_{i+1}^{t+1}\right]^{k} + c(\left[Q_{i+1}^{t+1}\right]^{k})^{b} - \frac{\Delta t}{\Delta x}Q_{i}^{t+1} - c(Q_{i+1}^{t})^{b} - q_{i+1}^{t+1}\Delta t}{\frac{\Delta t}{\Delta x} + cb(\left[Q_{i+1}^{t+1}\right]^{k})^{b-1}} \\ c = (\dfrac{1}{3600}n\chi^{\frac{2}{3}}S_f^{-\frac{1}{2}})^{\frac{3}{5}} \end{cases}$$

(11)

where $Q$ [L/s] is the water flow in surface river channel units, $A$ [dm$^2$] is the discharge
section area, $c$ is a custom intermediate variable, and $\chi$ [dm] is the wetted perimeter of the
discharge section area.
The underground runoff in the model includes the confluence of epikarst zone and



underground river. In the epikarst zone, the karst water-bearing media are highly
heterogeneous. For example, the crisscrossed karst fissure systems and conduit systems
consisted of large corrosion fractures. When rainfall infiltrates into the epikarst zone, water
moves slowly through the small karst fissure systems, while it flows rapidly in larger
conduits. The key to determining the confluence velocity lies in the width of karst fractures.
In the KHRUs (Fig. 3), the 10-cm width of the fracture was used as a threshold value
(Atkinson, 1977), meaning that if the fracture width exceeded 10 cm, then the water
movement into it was defined as rapid flow; otherwise, it was defined as slow flow. The
confluence in the epikarst zone was calculated by the following equation (modified from
Beven and Binley, 2006):

$$Q(t)_{ijk} = b_{ijk} \cdot \frac{\Delta h}{\Delta l} R_i C_j \cdot T(t)_{\text{slow/rapid}} \tag{12}$$

where

$$
\begin{cases}
T(t)_{\text{slow}} = nr \dfrac{\rho g R_i C_j L_k}{12v} \\[2mm]
T(t)_{\text{rapid}} = \dfrac{K_{ij}\left(e^{-f_{ij}h_{ij}} - e^{-f_{ij}z_{ij}}\right)}{f_{ij}}
\end{cases}
\tag{13}
$$

Here, $Q(t)_{ijk}$ [L/s] is the flow confluence in the epikarst zone at time $t$, $b_{ijk}$ [dm] is
the runoff width, $\dfrac{\Delta h}{\Delta l}$ is the dimensionless hydraulic gradient, $T(t)_{\text{slow/rapid}}$ is the
dimensionless hydraulic conductivity, $\rho$ [g/L] is the density of the water flow, $g$ [m/s²] is
gravitational acceleration, $n$ is the valid computational units, $R_i C_j L_k$ [L] is the volume of
the $ijk$-th KHRU, $v$ is kinematic viscosity coefficient, $f_{ij}$ is the attenuation coefficient in
the epikarst zone, $h_{ij}$ [dm] is the depth of shallow groundwater, and $z_{ij}$ [dm] is thickness of
epikarst zone.
The distinction between rapid and slow flows in the epikarst zone is not absolute. The
10-cm width of a karst fracture as the dividing threshold also has some subjectivity. In fact,
there is usually water exchange between the rapid and slow flows at the junction of large and
small fissures in karst aquifers. In the QMG model, this water exchange can be described



with this equation (modified form Li et al., 2021):

$$
\begin{cases}
Q = \alpha_{i,j,k}\left(h_n - h_{i,j,k}\right) \\
\alpha_{i,j,k} = \displaystyle\sum_{ip=1}^{np} \frac{\left(K_w\right)_{i,j,k}\,\pi d_{ip}\,\dfrac{1}{2}\left(\Delta l_{ip}\tau_{ip}\right)}{r_{ip}}
\end{cases}
\tag{14}
$$


Here, $\alpha_{i,j,k}$ [dm²/s] is the water exchange coefficient in the $ijk$-th KHRU,
$\left(h_n - h_{i,j,k}\right)$ [dm] is the water head difference between the rapid and slow flows at the
junction of large and small fissures in KHRUs, $np$ is the number of fissure systems
connected to the adjacent conduit systems, $\left(K_w\right)_{i,j,k}$ [dm/s] is the permeability coefficient
at the junction of a fissure and conduit, $d_{ip}$ and $r_{ip}$ [dm] are the conduit diameter and
radius, respectively, $\Delta l_{ip}$ [dm] is the length of the connection between conduits $i$ and $p$, and
$\tau_{ip}$ is the conduit curvature. Some of the parameters in this equation, such as $\left(K_w\right)_{i,j,k}$
and $\left(h_n - h_{i,j,k}\right)$, were obtained by conducting an infiltration test in the study area.
The confluence of the underground river system plays an important role for the
confluence at the basin outlet. To facilitate the calculation of confluence in the QMG model,
the underground river systems can be generalized into large multiple conduit systems.
During floods, these conduit systems are mostly under pressure. Whether the water flow is
laminar or turbulent depends on the flow regime at that time. The water flow into these
conduits is calculated by the Hagen–Poiseuille equation and the Darcy–Weisbach equation
(Shoemaker et al., 2008):

$$
\begin{cases}
Q_{\text{laminar}} = -A\dfrac{g d^2 \partial h}{32\nu \partial x} = -A\dfrac{\rho g d^2 \Delta h}{32\mu\tau\Delta l} \\[2ex]
Q_{\text{turbulent}} = -2A\sqrt{\dfrac{2gd\,|\Delta h|}{\Delta l\tau}}\log\left(\dfrac{H_c}{3.71d} + \dfrac{2.51\nu}{d\sqrt{\dfrac{2gd^3\,|\Delta h|}{\Delta l\tau}}}\right)\dfrac{\Delta h}{|\Delta h|}
\end{cases}
\tag{15}
$$


Here, $Q_{\text{laminar}}$ [L/s] is the water flow of the laminar flow in the conduit systems, $A$



[dm$^2$] is the conduit cross-sectional area, $d$ [dm] is the conduit diameter, $\rho$ [kg/dm$^3$] is the
density of the underground river, $\nu = \mu / \rho$ is the coefficient of kinematic viscosity,
$\Delta h / \tau \Delta l$ is the hydraulic slope of the conduits, $\tau$ is the dimensionless conduit curvature,
$Q_{\text{turbulent}}$ [L/s] is the turbulent flow in the conduit systems, and $H_c$ [dm] is the average
conduit wall height.

**3.2. Parameter Optimization**

In total, the QMG model has 12 parameters, of which flow direction and slope are

topographic parameters that can be determined from the DEM without parametric
optimization, while the remaining 10 parameters require calibration. Other distributed
hydrological models with multiple structures usually have many parameters. For example,
the Karst-Liuxihe model (Li et al., 2021) has 15 parameters that need to be calibrated. In the
QMG model, each parameter is normalized as

$$x_i = x^*_i / x_{i0}, \tag{16}$$

where $x_i$ is the dimensionless parameter value $i$ after it is normalized, $x^*_i$ is the

parameter value $i$ with an actual physical property, and $x_{i0}$ is the initial or final value of $x_i$.
Through the processing of equation (16), the value range of the model parameters is limited to
a hypercube $K_n = (X \mid 0 \le x_i \le 1, \ i = 1, 2, ..., n)$, which is a dimensionless value. Such a
normalized treatment can ignore the influence of the spatiotemporal variation of the
underlying surface attributes on the parameters and at the same time simplify the parameter
classification and parameter number of the model to a certain extent. Accordingly, the model
parameters can be further divided into rainfall-evaporation parameters, epikarst-zone
parameters, and underground-river parameters. Table 1 lists the parameters of the QMG
model.

Table 1 Parameters of QMG model.

Because the QMG model has relatively few parameters, it is possible to calibrate

them manually, which has the advantage that the operation is easy to implement and does
not require a special program for parameter optimization. However, the disadvantage is that





it is subjective, which can lead to great uncertainty in the manual parameter calibration process. To compare the effects of parameter optimization on model performance, this study used both a manual calibration of parameters and the improved chaotic particle swarm algorithm (IPSO) for the automatic calibration of model parameters, and compared the effects of both on flood simulation.

In general, the structure and parameters of a standard particle swarm algorithm (PSO) are simple, with the initial parameter values obtained at random. For parameter optimization in high-dimensional, multi-peak hydrological models, the standard PSO is easily limited to a local convergence and cannot achieve the optimal effect, while the late evolution of the algorithm may also cause problems, such as precocity and stagnant evolution, due to the "inert" aggregation of particles, which seriously affects the efficiency of parameter selection. It is necessary to overcome the above problems and make the algorithm converge to the global optimal solution with a high probability. In parameter optimization for the QMG model, we improved the standard PSO algorithm by adding chaos theory, and developed the IPSO, where 10 cycles of chaotic disturbances were added to improve the activity of the particles. The inverse mapping equation of the chaotic variable is as follows:

$$\begin{cases} X_{ij} = X_{\min} + (X_{\max} - X_{\min}) * Z_{ij} \\ Z'_{ij} = (1-\alpha)Z^* + \alpha Z_{ij} \end{cases} \tag{17}$$

where $X_{ij}$ is the optimization variable for the model parameters, $(X_{\max} - X_{\min})$ is the difference between its maximum and its minimum, $Z_{ij}$ is the variable before the disturbance is added, and $Z'_{ij}$ represents the chaotic variables after a disturbance is added, $\alpha$ is a variable determined by the adaptive algorithm, $0 \le \alpha \le 1$, and $Z^*$ is the chaotic variable formed when the optimal particle maps to the interval [0,1]. In parameter optimization, the flowchart of the IPSO is shown in Fig. 4.

Figure 4 Algorithm flow chart of the IPSO.



**3.3. Uncertainty Analysis**

As a type of mathematical and physical model, a hydrological model has some uncertainty in flood simulation and forecasting because of the errors in system structure and the algorithm (Krzysztofowicz and Kelly, 2000). A multi-parametric sensitivity analysis method (Choi et al., 1999; Li et al., 2020) was used to analyze the sensitivity of the parameters in the QMG model. Parameter sensitivity analysis steps are as follows.

1) Selection of appropriate objective function

The Nash–Sutcliffe coefficient is widely used as the objective function to evaluate the performance of hydrological models (Li et al., 2020, 2021). It was therefore used to assess the QMG model. Because the most important factor in flood forecasting is the peak discharge, it is used in the Nash coefficient equation:

$$NSC = 1 - \frac{\sum_{i=1}^{n}\left(Q_i - Q_i^{'}\right)^2}{\sum_{i=1}^{n}\left(Q_i - \overline{Q}\right)^2}, \tag{18}$$

where $NSC$ is Nash–Sutcliffe coefficient values, $Q_i$ [L/s] is the observed flow discharges, $Q_i'$ [L/s] is the simulated discharge, $\overline{Q}$ [L/s] is the average value of the observed discharges, and $n$ [h] is the observation period.

2) Parameter sequence sampling

The Monte Carlo sampling method was applied to sample 8,000 groups of parameter sequences. The parametric sensitivity of the QMG model was analyzed and evaluated by comparing the differences between the a priori and a posteriori distributions of the parameters.

3) Parameter sensitivity assessment

The a priori distribution maens its probability distribution of a model parameter, while the a posteriori distribution can be calculated based on the simulation result of the parametric optimization. If there is a significant difference between them, then the parameter being tested has a high sensitivity, whereas if there is no obvious difference, then the parameter is insensitive. The parametric priori distribution is calculated as follows:





$$\begin{cases} P_{i,j}(NSC_{i,j} \geq 0.85) = \dfrac{n}{N+1} \times 100 \\ \sigma_i = \displaystyle\sum_{j=1}^{n} \left( P_{i,j} - \overline{P_{i,j}} \right)^2 \end{cases}$$
(19)

where $P_{i,j}$ is the priori distribution'probability when $NSC_{i,j} \geq 0.85$. We used a
simulated Nash coefficient of 0.85 as the threshold value, and $n$ was the number of
occurrences of a Nash coefficient greater than 0.85 in flood simulations. In each simulation,
only a certain parameter was changed, while the remaining parameters remained unchanged.
If the Nash coefficient of this simulation exceeded 0.85, then the flood simulation results
were considered acceptable. The term $\sigma_i$ is the difference between the acceptable value
and its mean, which represents the parametric sensitivity ($0 < \sigma_i < 1$). The higher the $\sigma_i$
value, the more sensitive the parameter. $N$ is the 8,000 parameter sequences, and $\overline{P_{i,j}}$ is the
average value of the a priori distribution.
**3.4. Model Setting**
Once the model was built, some of the initial conditions had to be set before running it
to simulate and forecast floods, such as basin division, the setting of initial soil moisture, and
the assumption of the initial parameter range. 1) In the study area, the entire Qingmuguan
karst basin was divided into 893 KHRUs, including 65 surface river units, 466 hill slope
units, and 362 underground river units. The division of these units formed the basis for
calculating the process of runoff generation and convergence. 2) The initial soil moisture
was set to 0–100% of the saturation moisture content in the basin, and the specific soil
moisture before each flood had to be determined by a trial calculation. 3) The waterhead
boundary conditions of the groundwater were determined by a tracer test in the basin, where
a perennial stable water level adjacent the groundwater-divide was used as the fixed
waterhead boundary. The base flow of the underground river was determined to be 35 L/s
from the perennial average dry season runoff. 4) The range of initial parameters and
convergence conditions were assumed before parameter optimization (Figure 4). 5)
Parameter optimization and flood simulation validated the performance of the QMG model





in karst basins.
**4. Results and Discussion**
**4.1. Parameter Sensitivity Results**

The number of parameters in a distributed hydrological model is generally large, and it

is important to perform a sensitivity analysis of each parameter to quantitatively assess the
impact of the different parameters on model performance. In the QMG model, each
parameter was divided into four categories according to its sensitivity: (i) highly sensitive,
(ii) sensitive, (iii) moderately sensitive, and (v) insensitive. In the calibration of model
parameters, insensitive ones do not need to be calibrated, which can greatly reduce the
amount of calculation and improve the efficiency of model operation.

The flow process in the calibration period (14 April to 10 May 2017) was adopted to

calculate the sensitivity of the model parameters, for which the calculation principle was
equation (19), and the parameter sensitivity results are calculated in Table 2.

Table 2 Parametric sensitivity results in QMG model.

In Table 2, the value of $\sigma_i$ [equation (19)] represents a parameter's sensitivity, and the

higher the value, the more sensitive the parameter is. From the results in Table 2, it was
found that the rainfall infiltration coefficient, rock permeability coefficient, rock porosity,
and the related parameters of soil water content, such as the saturated water content, and
field capacity, were sensitive parameters. The order of parameter sensitivity was as follows:
infiltration coefficient > permeability coefficient > rock porosity > specific yield > saturated
water content > field capacity > flow direction > thickness > slope > Soil coefficient >
channel roughness > evaporation coefficient.

In the QMG model, parameters are classified as highly sensitive, sensitive, moderately

sensitive, and insensitive according to their influence on the flood simulation results. In
Table 4, we divided the sensitivity of model parameters into four levels based on the $\sigma_i$
value: 1) highly sensitive parameters, $0.8 < \sigma_i < 1$; 2) sensitive parameters, $0.65 < \sigma_i < 0.8$;





3) moderately sensitive parameters, $0.45 < \sigma_i < 0.65$; and 4) insensitive parameters,
$0 < \sigma_i < 0.45$. The highly sensitive parameters were the infiltration coefficient, permeability
coefficient, rock porosity, and specific yield. The sensitive parameters were the saturated
water content, field capacity, and thickness of the epikarst zone. The moderately sensitive
parameters were flow direction, slope, and soil coefficient. The insensitive parameters were
channel roughness and the evaporation coefficient.
**4.2. Parametric Optimization**
In total, the QMG model has 12 parameters, of which only eight need to be optimized,
which is relatively few for distributed models. The parameters of flow direction and slope as
well as the insensitive parameters of channel roughness and the evaporation coefficient need
not be calibrated, which can improve the convergence efficiency of the model parameter
optimization.
In the study area, 18 karst floods during the period of 14 April 2017 to 10 June 2019
were recorded at underground river outlet to validate the effects of the QMG model in karst
hydrological simulations. The calibration period was 14 April to 10 May 2017 at the
beginning of the flow process, with the remainder of the time being the validation period. In
the QMG model, the IPSO algorithm was used to optimize the model parameters. To show
the necessity of parameter optimization for the distributed hydrological model, the study
specifically compared the flood simulations obtained using the initial parameters of the
model (without parameter calibration) and the optimized parameters. Fig. 5 shows the
iteration process of parameter optimization for the QMG model.

Figure 5 Iteration process of parametric optimization.

Fig. 5 shows that almost all parameters fluctuated widely at the beginning of the
optimization, and then after about 15 iterations of the optimization calculation, most of the
linear fluctuations become significantly less volatile, which indicated that the algorithm
tended to converge (possibly only locally). When the number of iterations exceeded 25, all
parameters remained essentially unchanged, meaning that the algorithm had converged (at
this point there was global convergence). It took only 25 iterations to reach a definite





convergence of the parameter rates with this IPSO algorithm, which is extremely efficient in
terms of the parameter optimization of distributed hydrological models. In previous studies
of the parametric optimization for the Karst-Liuxihe model in similar basin areas, 50
automatic parameter optimization iterations were required to reach convergence (Li et al.,
2021), demonstrating the effectiveness of the IPSO algorithm.
To evaluate the effect of parameter optimization, the convergence efficiency of the
algorithm, and more importantly, the parameters after calibration were used to simulate
floods. Fig. 6 shows the flood simulation effects.
Figure 6 Flow simulation results of QMG model based on parameter optimization.
Fig. 6 shows that the flows simulated by parameter optimization were better than those
simulated by the initial model parameters. The simulated flow processes based on the initial
parameters were relatively small, with the simulated peak flows in particular being smaller
than the observed values, and there were large errors between the two values. In contrast, the
simulated flows produced by the QMG model after parameter optimization were very
similar to the observed values, which indicates that calibration of the model parameters is
necessary and that there was an improvement in parameter optimization through the use of
the IPSO algorithm in this study. In addition, it was found that the flow simulation effect
was better in the calibration periods than in the validation periods (Fig. 6).
To compare the results of the flow processes simulation with the initial model
parameters and the optimized parameters, six evaluation indices (Nash–Sutcliffe coefficient,
correlation coefficient, relative flow process error, flood peak error, water balance
coefficient, and peak time error) were applied in this study, and the results are presented in
Table 3.

Table 3 Flood simulation evaluation index through parametric optimization.

Table 3 shows that the evaluation indices of the flood simulations after parametric
optimization were better than those of the initial model parameters. The average values of
the initial parameters for these six indices 0.81, 0.74, 27%, 31%, 0.80, and 5 h, respectively.
For the optimized parameters, the average values were 0.90, 0.91, 16%, 14%, 0.94, and 3 h,



respectively. The flood simulation effects after parameter optimization clearly improved,
implying that parameter optimization for the QMG model is necessary, and the IPSO
algorithm for parameter optimization is an effective approach that can greatly improve the
convergence efficiency of parameter optimization, and also ensure that the model performs
well in flood simulations.
**4.3. Model Validation in Flood Simulations**
Following parameter optimization, we simulated the whole flow process (14 April 2017
to 10 June 2019 ) based on the optimized and initial parameters of the QMG model (Fig. 6),
which enabled a visual reflection of the model used in the simulation of a long series of flow
processes. To reflect the simulation effect of the model for different flood events, we divided
the whole flow process into 18 flood events, then used the initial parameters of the model
and the optimized parameters, respectively, to verify the model performance in flood
simulations. Fig. 7 and Table 4 show the flood simulation effects and their evaluation indices
using both the initial and the optimized parameters.
Figure 7 Flood simulation effects based on initial and optimized parameters.
Table 4 Flood simulation indices for model validation.
Fig. 7 shows that the flood simulation results using the initial parameters were smaller
than the observed values, and the model performance improved in flood simulations after
parameter optimization. The simulated flood processes were in good agreement with
observations, and were especially effective for simulating flood peak flows. From flood
simulation indices in Table 4, the average water balance coefficient based on the initial
parameters was 0.69, i.e., much less than 1, indicating that the simulated water in the model
was unbalanced. After parameter optimization, the average value was 0.92, indicating that
parameter optimization had a significant impact on the model water balance calculation.
Table 4 shows that the average values of the six indices (Nash–Sutcliffe coefficient,
correlation coefficient, relative flow process error, flood peak error, water balance
coefficient, and peak time error) for the initial parameters were 0.79, 0.74, 26%, 25%, 0.69,
and 5 h, respectively, while for the optimized parameters the average values were 0.92, 0.90,
10%, 11%, 0.92, and 2 h, respectively. All evaluation indices improved after parameter
optimization, with the average values of the Nash coefficient, correlation coefficient, and
water balance coefficient increasing by 0.13, 0.16, and 0.23, respectively. The average
values of the relative flow process error, flood peak error, and peak time error decreased by
15%, 14%, and 3 h, respectively. These reasonable flood simulation results confirmed that
parameter optimization by the IPSO algorithm was necessary and effective for the QMG
model.
Compared with the overall flow process simulation shown in Figure 6, each flood
process was better simulated by the QMG model (Fig. 7). This was because in the function
of the QMG model and its algorithm design, the main consideration was the calculation of
the flood process, but the correlation algorithm of the dry season runoff was not described
well enough. For example, equations (12)–(15) are the flood convergence algorithm. As a
result, the model is not good at simulating other flow processes, such as dry season runoff,
leading to a low accuracy in the overall flow process. The next phase of our research will
focus on refining the algorithm related to dry season runoff and improving the
comprehensive performance of the model.
**4.4. Uncertainty Analysis**
**4.4.1 Assessment and Reduction of Uncertainty**
In general, the uncertainty in model simulations are mainly derived from three aspects
of the model: 1) the uncertainty of the model input data, 2) the uncertainty of model
structure and the algorithm, and 3) the uncertainty of model parameters. In the practical
application of a hydrological model, these three uncertainties are usually interwoven, which
leads to the overall uncertainty of the final simulation results (Krzysztofowicz, 2014).
Therefore, this study focused on the input data, the model structure, and parameter
uncertainty to reduce the overall uncertainty of simulation results.
First, the input data (e.g., rainfall, flood event, and some hydrogeological data) were
validated and pre-processed through observations in this study, which substantially reduced
the uncertainty.





Second, we simplified the structure of the QMG model. The model was designed with
full consideration of the relationship between the amount of data required to build the model
and its performance for flood simulation and forecasting in karst regions, and the model's
entire framework was integrated through simple structures and easy-to-implement
algorithms, using the concept of distributed hydrological modeling. Conventionally, the
extent of uncertainty is increased with the growing complexity of the model structure. We
therefore ensured that the structure of the QMG model was simple when it was designed,
and the model was divided into surface and underground double layer structures. Such a
relatively simple structure made the uncertainty of the model structure reduced largely. In
contrast, the underground structure of our previous Karst-Liuxihe model (Li et al., 2021) has
five layers, which leads to great uncertainty.
Third, appropriate algorithms for runoff generation and confluence were selected. The
design function of different models is different, which leads to great differences in the
algorithms used. In the QMG model, most of the rainfall-runoff algorithms used have been
validated by the research results of others, and some of them were improved to suit karst
flood simulation and forecasting by the QMG model. For example, the algorithm of excess
infiltration runoff generation [equation (2)] was an improvement of the version used in the
Liuxihe model (Chen, 2009, 2018; Li et al., 2020).
Finally, the algorithm of parameter optimization was improved. Considering the
shortcomings of the standard PSO algorithm that tends to reach a local convergence, this
study developed the IPSO for parameter optimization by adding chaotic perturbation factors.
The flood simulation results after parameter optimization were much better than those of the
initial model parameters (Figs. 6 and 7 and Tables 2 and 3), which indicates that parameter
optimization is necessary for a distributed hydrological model and can reduce the
uncertainty of model parameters.
**4.4.2 Parameter Sensitivity Analysis**
From the results of parameter sensitivity in Table 2, it can be seen that the rainfall
infiltration coefficient in the QMG model was the most sensitive parameter. It was the key to





determining the generation of excess infiltration surface runoff and dividing surface runoff
from subsurface runoff. If the rainfall infiltration coefficient was greater than the infiltration
capacity, it would generate excess infiltration surface runoff on the exposed karst landforms;
otherwise, all rainfall would infiltrate to meet the water deficit in the vadose zone, and then
continue to seep down into the underground river system, eventually flowing out of the basin
through the underground river outlet. The confluence modes of surface runoff and
underground runoff were completely different, resulting in a large difference in the
simulated flow results. Therefore, the rainfall infiltration coefficient had the greatest impact
on the final flood simulation results.

Other highly sensitive parameters such as the rock permeability coefficient, rock

porosity, and specific yield were used as the basis for the division of slow flow in karst
fissures and rapid flow in conduits. The division of slow and rapid flows also had a great
impact on the discharge at the outlet of the basin. Slow flow plays an important role in water
storage in a karst aquifer and is very important for the replenishment of river base flow in
the dry season. Rapid flow in large conduit systems dominates of the flood runoff and is the
main component of the flood water volume in the flood season.

Parameters related to the soil water content, including the saturated water content, field

capacity, and thickness, were sensitive parameters and had a large influence on the flood
simulation results. This is because the soil moisture content prior to flooding affects how
flood flows rise and when peaks occur. If the soil is already very wet or even saturated
before the flooding, the flood will quickly rise to reach a peak, and the process line of the
flood peak flow will be sharp and thin. This type of flood process forms easily and can lead
to disaster-causing flood events. In contrast, if the soil in the basin is very dry before the
flooding, the rainfall will first meet the water shortage of the vadose zone, and after it is
replenished the rainfall will infiltrate into the underground river. The flood peak of the river
basin outlet is therefore delayed.

The moderately sensitive parameters were flow direction, slope, and the soil coefficient.

They had a specific influence on the flood simulation results, but the influence was not as
great as the highly sensitive and sensitive parameters. The insensitive parameters were





channel roughness and the evaporation coefficient. The amount of water lost by
evapotranspiration is very small in the total flood water, and it was therefore the least
sensitive parameter in the QMG model.

**5. Conclusions**

This study proposed a new distributed physically based hydrological model, i.e., the
QMG model, to simulate floods accurately in karst trough valley. The main conclusions of
this paper are as follows.
This QMG model has a high application potential in karst hydrology simulations. Other
distributed hydrological models usually have multiple structures, resulting in the need for a
large amount of data to build models in karst areas (Kraller et al., 2014). The QMG model
has only a double-layer structure, with a clear physical meaning, and a small amount of
basic data is needed to build the model in karst areas, such as some necessary
hydrogeological data. For example, the distribution and flow direction of underground rivers
is required, which can be inferred from a tracer test, leading to a low modeling cost. There
were fewer parameters in the QMG model than in other distributed hydrological models,
with only 10 parameters that needed to be calibrated.
The flood simulation after parameter optimization was much better than the simulation
using the initial model parameters. After parameter optimization, the average value of Nash
coefficient, correlation coefficient, and water balance coefficient increased by 0.13, 0.16,
and 0.23, respectively; while the average relative flow process error, flood peak error, and
peak time error decreased by 15%, 14%, and 3 h, respectively. Parameter optimization is
necessary for a distributed hydrological model, and the improvement of the IPSO algorithm
in this study was an effective way to achieve this.
In the QMG model, the rainfall infiltration coefficient, $I_c$, rock permeability coefficient,
$K$, rock porosity, $R_p$, and the parameters related to the soil water content were sensitive
parameters. The order of parameter sensitivity was: infiltration coefficient > permeability
coefficient > rock porosity > specific yield > saturated water content > field capacity > flow
direction > thickness > slope > soil coefficient > channel roughness > evaporation



coefficient.
This QMG model is suitable for karst trough valley basins, where the topography is
conducive to the spread of flood water. Whether this model is applicable to other karst areas
in non-trough valley regions still needs to be verified in the future studies. In addition, the
basin area is very small, where the hydrological similarity between different small basin
areas varies greatly (Kong and Rui, 2003). The size of the area to be modeled has a great
influence on the choice of model spatial resolution (Chen et al., 2017). Therefore, whether
the QMG model is suitable for flood forecasting in large karst basins needs to be
determined.
**Model development.**
This QMG model presented in this study uses the Visual Basic language programming. The
general framework of the model and the algorithm consist of three parts: the modeling
approach, the algorithm of rainfall-runoff generation and confluence, and the parameter
optimization algorithm. As a free and open source hydrological modeling program (QMG
model-V1.0), we provide all modeling packages, including model code, installation package,
simulation data package and user manual, free of charge. It is important to note that the
model we provide are for scientific research purposes only and should not be used for any
commercial purposes. Creative Commons Attribution 4.0 International.
Model installation program can be downloaded from ZENODO, cite as JI LI. (2021, June
16). QMG model-V1.0. Zenodo. http://doi.org/10.5281/zenodo.4964701, and
http://doi.org/10.5281/zenodo.4964697 (registration required). User manual can be
downloaded from http://doi.org/10.5281/zenodo.4964754.
**Code availability.**
All code for the QMG model-V1.0 in this paper are available and free, the code can be
downloaded from ZENODO, Cite as JI LI. (2021, June 16). QMG model-V1.0 code
(Version v1.0). Zenodo. http://doi.org/10.5281/zenodo.4964709 (registration required).
**Data availability.**
All data used in this paper are available, findable, accessible, interoperable, and reusable.
The simulation data and modelling data package can be downloaded from
http://doi.org/10.5281/zenodo.4964727. The DEM was downloaded from the Shuttle Radar
Topography Mission database at http://srtm.csi.cgiar.org. The land use-type data were
downloaded from http://landcover.usgs.gov, and the soil-type data were downloaded from
http://www.isric.org. These data were last accessed on 15 October 2020.
**Author contributions.** JIL was responsible for the calculations and writing of the whole



paper. DY and YJ helped conceive the structure of the model. ZF and JL provided significant
assistance in the English translation of the paper. MM provided flow data of the study area.
**Competing interests.**
The authors declare that they have no conflicts of interest.
**Acknowledgments.**
This study was supported by the National Natural Science Foundation of China (41830648),
Chongqing Municipal Science and Technology Commission Fellowship Fund
(cstc2019yszx-jcyjX0002), and the drought monitoring, analysising and early warning of
typical prone-to-drought areas of Chongqing (20C00183).



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





**Tables**
Table 1 Parameters of the QMG model.

| Parameters | Variable name | Physical property |
|---|---|---|
| Infiltration coefficient | $I_c$ | Meteorology |
| Evaporation coefficient | $\lambda$ | Vegetation cover |
| Soil thickness | $h$ | Karst aquifer |
| Soil coefficient | $S_b$ | Soil type |
| Saturated water content | $S_c$ | Soil type |
| Rock porosity | $R_p$ | Karst aquifer |
| Field capacity | $F_c$ | Soil type |
| Permeability coefficient | $K$ | Karst aquifer |
| Flow direction | $F_d$ | Landform |
| Slope | $S_0$ | Landform |
| Specific yield | $S_y$ | Karst aquifer |
| Channel roughness | $n$ | Landform |

Table 2 Parametric sensitivity results in QMG model.

| Infiltration coefficient/$I_c$ | Evaporation coefficient/$\lambda$ | Thickness/$h$ | Soil coefficient/$S_b$ | Saturated water content/$S_c$ | Specific yield/$S_y$ |
|---|---|---|---|---|---|
| 0.92 | 0.24 | 0.71 | 0.58 | 0.8 | 0.83 |

| Flow direction/$F_d$ | Slope/$S_0$ | Rock porosity/$R_p$ | Field capacity/$F_c$ | Permeability coefficient/$K$ | Channel roughness/$n$ |
|---|---|---|---|---|---|
| 0.74 | 0.68 | 0.86 | 0.78 | 0.89 | 0.36 |

Table 3 Flood simulation evaluation index through parametric optimization.

| Parameter optimization | Parameter types | Nash coefficient | Correlation coefficient | Relative flow process error/% | Flood peak error/% | Water balance coefficient | Peak time error (hours) |
|---|---|---|---|---|---|---|---|
| calibration periods | initial | 0.82 | 0.77 | 24 | 29 | 0.82 | 4 |
| | optimized | 0.91 | 0.94 | 14 | 12 | 0.95 | 2 |
| validation periods | initial | 0.79 | 0.71 | 29 | 32 | 0.77 | 6 |
| | optimized | 0.88 | 0.87 | 18 | 16 | 0.92 | 3 |
| average value | initial | 0.81 | 0.74 | 27 | 31 | 0.80 | 5 |
| | optimized | 0.90 | 0.91 | 16 | 14 | 0.94 | 3 |





Table 4 Flood simulation indices for model validation.

| Floods | Parameter types | Nash coefficient | Correlation coefficient | Relative flow process error/% | Flood peak error/% | Water balance coefficient | Peak time error/ (hours) |
|---|---|---|---|---|---|---|---|
| 2017042408 | initial | 0.77 | 0.7 | 28 | 29 | 0.71 | -5 |
|  | optimized | 0.95 | 0.89 | 11 | 15 | 0.88 | -2 |
| 2017050816 | initial | 0.78 | 0.71 | 19 | 19 | 0.76 | -4 |
|  | optimized | 0.92 | 0.88 | 11 | 9 | 0.94 | -2 |
| 2017061518 | initial | 0.76 | 0.6 | 25 | 32 | 0.63 | -5 |
|  | optimized | 0.91 | 0.93 | 12 | 11 | 0.95 | -3 |
| 2017071015 | initial | 0.78 | 0.82 | 25 | 37 | 0.64 | -4 |
|  | optimized | 0.92 | 0.87 | 8 | 7 | 0.94 | -2 |
| 2017091512 | initial | 0.81 | 0.62 | 21 | 16 | 0.78 | -5 |
|  | optimized | 0.9 | 0.92 | 13 | 10 | 0.9 | -4 |
| 2017100815 | initial | 0.75 | 0.68 | 30 | 26 | 0.62 | -2 |
|  | optimized | 0.94 | 0.86 | 11 | 15 | 0.92 | -1 |
| 2018052016 | initial | 0.78 | 0.68 | 25 | 21 | 0.67 | 5 |
|  | optimized | 0.91 | 0.93 | 10 | 13 | 0.94 | 2 |
| 2018060815 | initial | 0.82 | 0.79 | 27 | 22 | 0.69 | -6 |
|  | optimized | 0.9 | 0.92 | 11 | 12 | 0.93 | -4 |
| 2018071212 | initial | 0.84 | 0.75 | 26 | 24 | 0.61 | 5 |
|  | optimized | 0.91 | 0.88 | 8 | 15 | 0.92 | 3 |
| 2018081512 | initial | 0.71 | 0.78 | 26 | 24 | 0.78 | -4 |
|  | optimized | 0.89 | 0.94 | 12 | 11 | 0.89 | -3 |
| 2018090516 | initial | 0.85 | 0.68 | 28 | 23 | 0.68 | -5 |
|  | optimized | 0.93 | 0.87 | 12 | 10 | 0.92 | -2 |
| 2018092514 | initial | 0.79 | 0.78 | 23 | 19 | 0.59 | 5 |
|  | optimized | 0.88 | 0.88 | 9 | 11 | 0.89 | 2 |
| 2018101208 | initial | 0.78 | 0.81 | 28 | 25 | 0.63 | 5 |
|  | optimized | 0.92 | 0.94 | 11 | 10 | 0.94 | 2 |
| 2018111208 | initial | 0.79 | 0.81 | 25 | 24 | 0.65 | -6 |
|  | optimized | 0.94 | 0.86 | 13 | 12 | 0.92 | -2 |
| 2019042512 | initial | 0.78 | 0.8 | 26 | 36 | 0.8 | 5 |
|  | optimized | 0.89 | 0.94 | 9 | 16 | 0.93 | 2 |
| 2019051513 | initial | 0.84 | 0.77 | 32 | 27 | 0.79 | 4 |
|  | optimized | 0.91 | 0.88 | 9 | 13 | 0.95 | 2 |
| 2019052516 | initial | 0.74 | 0.75 | 29 | 26 | 0.63 | -5 |
|  | optimized | 0.92 | 0.86 | 7 | 15 | 0.96 | -2 |
| 2019060518 | initial | 0.85 | 0.83 | 28 | 25 | 0.78 | -4 |
|  | optimized | 0.95 | 0.96 | 10 | 12 | 0.92 | -2 |
| average value | initial | 0.79 | 0.74 | 26 | 25 | 0.69 | 5 |
|  | optimized | 0.92 | 0.9 | 10 | 11 | 0.92 | 2 |



**Figures**

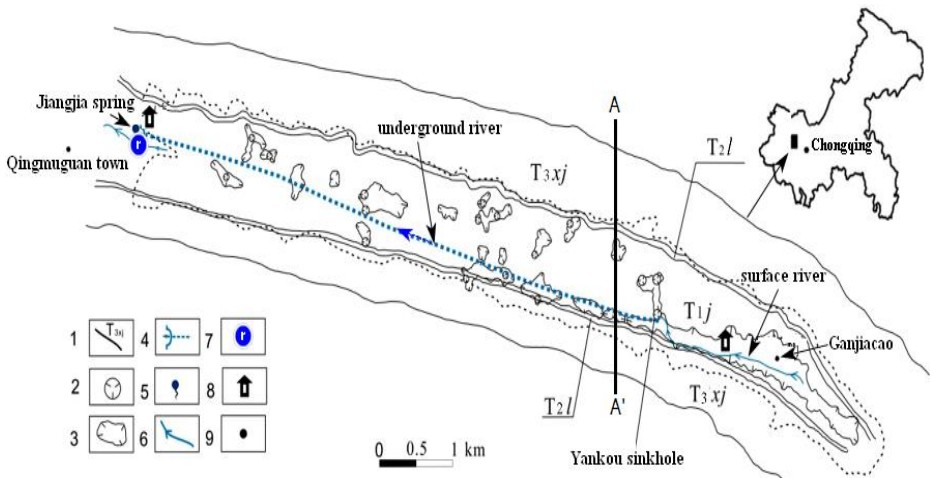

1- stratigraphic boundary, 2-sinkhole, 3- karst depression, 4- underground river, 5-karst spring, 6-surface river,7-river gauge, 8- rain gauge, and 9- geographical name

a.    Qingmuguan karst basin (modified from Yu et al.,2016)

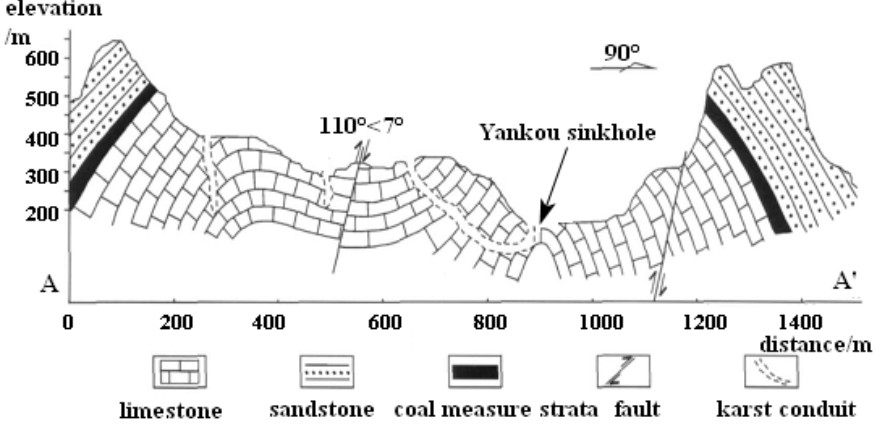

b. Lithologic cross section of Yankou sinkhole/AA'(modified from Zhang,2012)



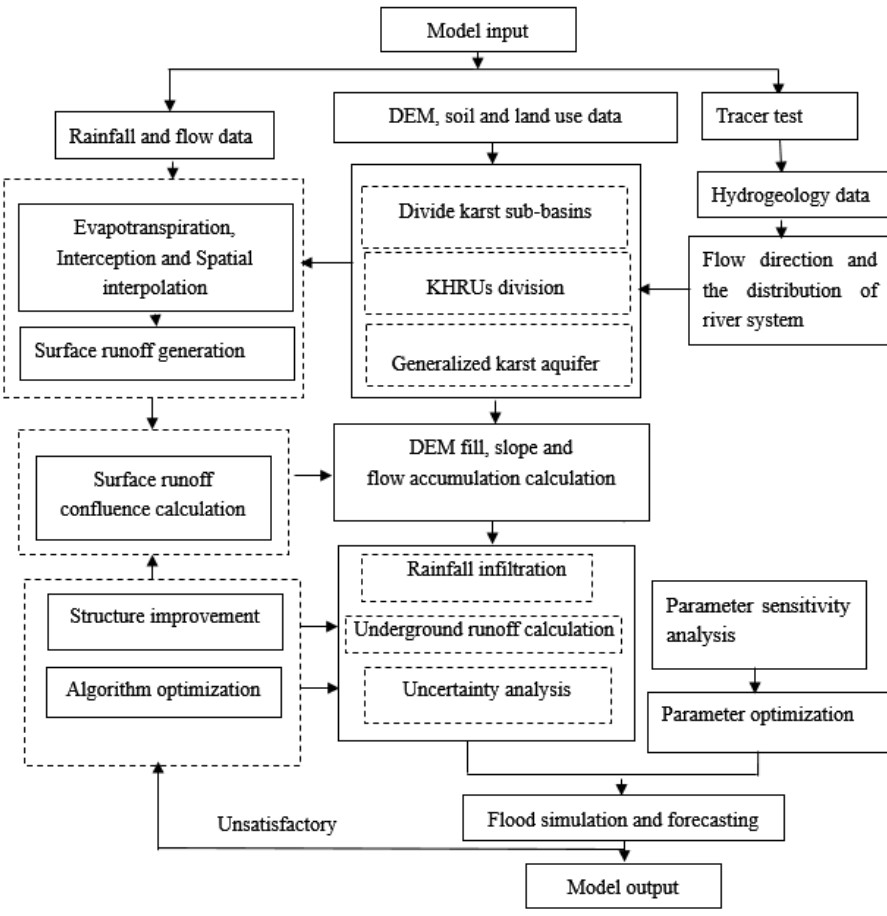


c. Longitudinal profile of the study area (modified from Yang et al.,2008)

Figure 1 The Qingmuguan karst basin.


Figure 2 Modeling flow chart of QMG (Qingmuguan) model.





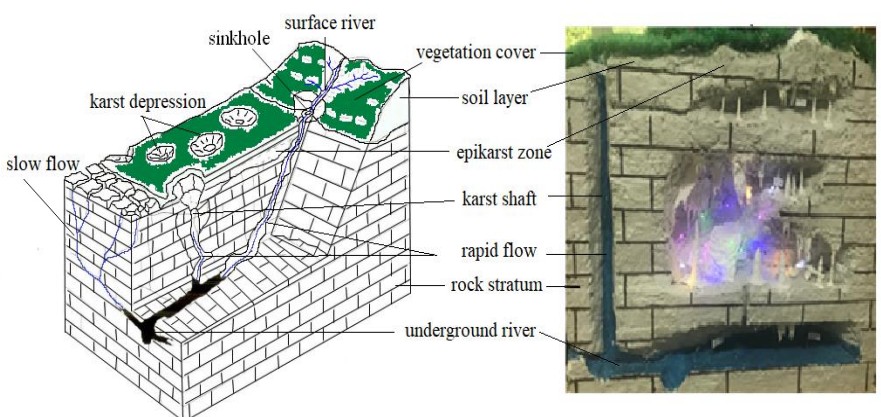


Figure 3 Spatial structure of the KHRUs (Li et al.,2021).

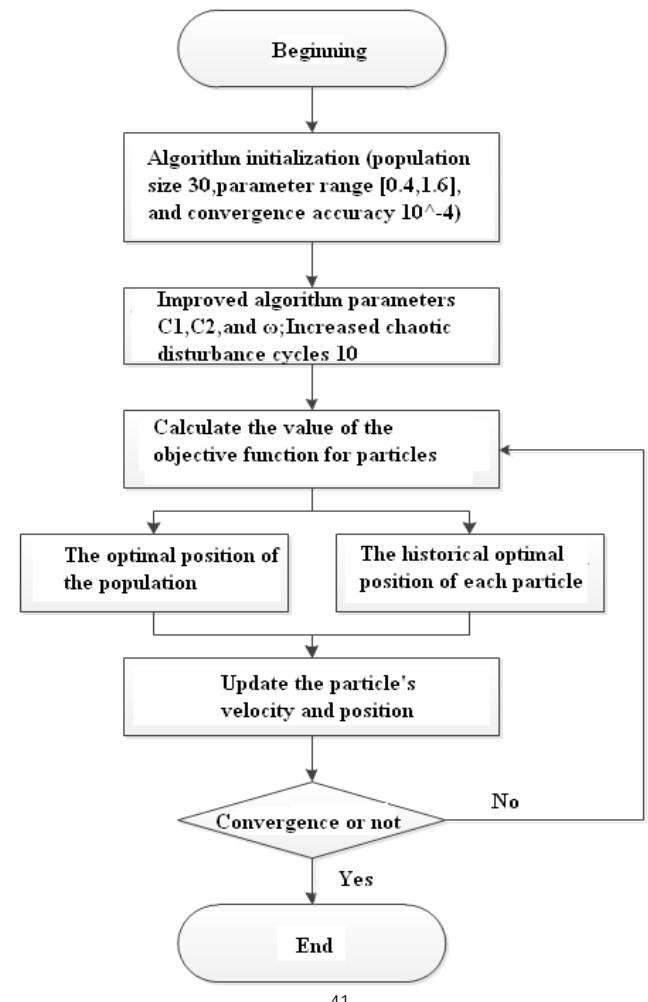






Figure 4 Algorithm flow chart of the IPSO.

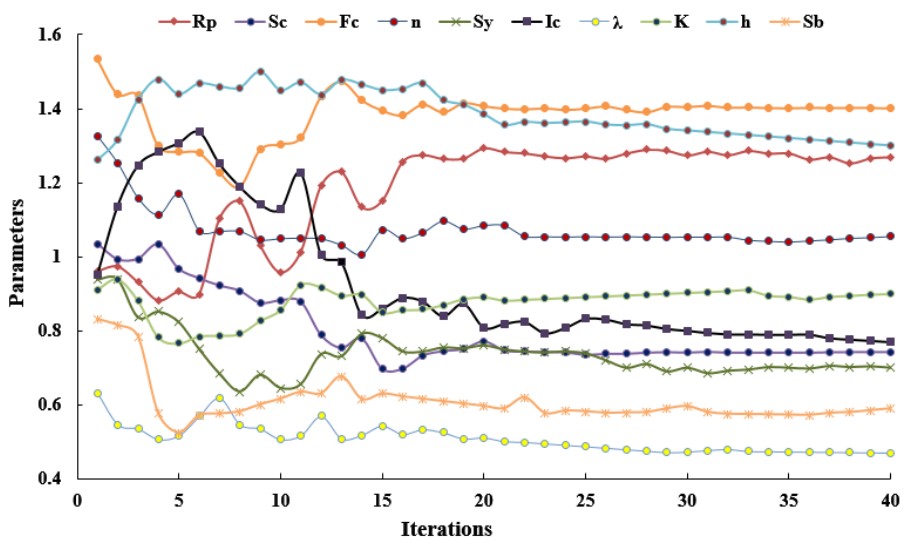


Figure 5 Iteration process of parametric optimization.

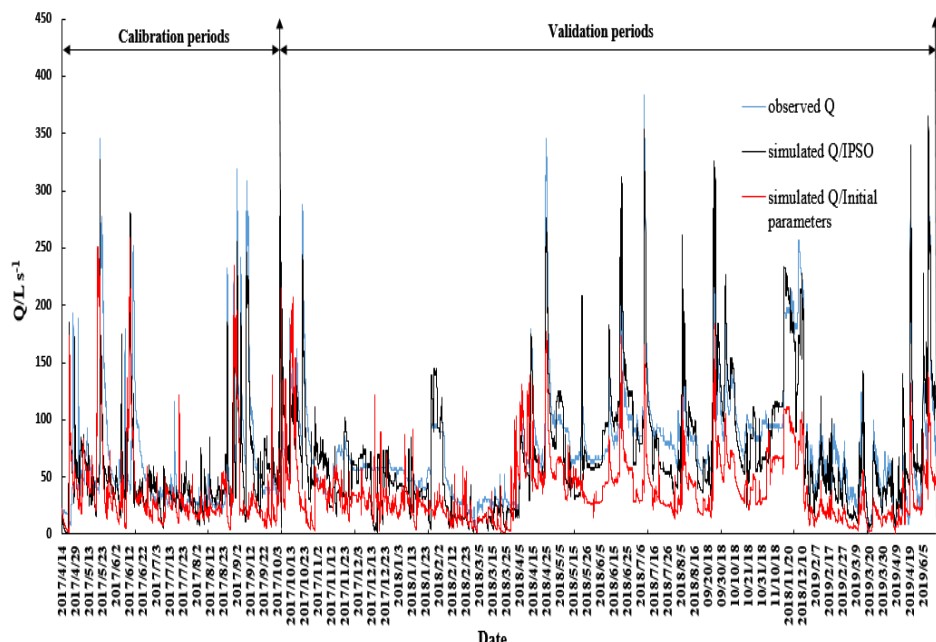


Figure 6 Flow simulation results of QMG model based on parameter optimization.




a. flood 201704240800             b. flood 201705081600


c.flood 201706151800             d. flood 201707101530


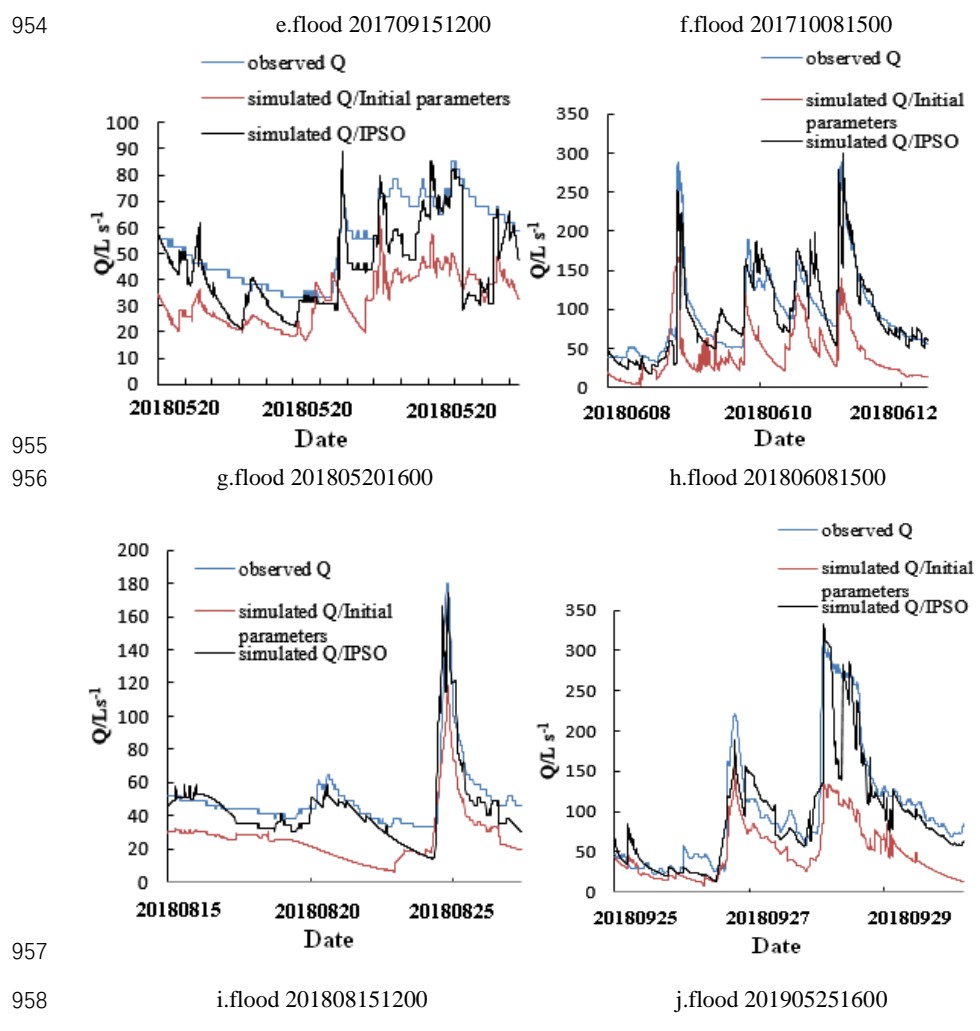






Figure 7 Flood simulation effects based on initial and optimized parameters.