# Peer review of "A physically based distributed karst hydrological model (QMG model-V1.0) for flood simulations"

_Geoscientific Model Development, 2021_

## Referee Comment (RC2)

[revised manuscript text omitted]

               Figure 4 Algorithm flow chart of the IPSO.

[Figure]

               Figure 5 Iteration process of parametric optimization.

[Figure]

 Figure 6 Flow simulation results of QMG model based on parameter optimization.

[Figure]

a. flood 201704240800        b. flood 201705081600

c.flood 201706151800        d. flood 201707101530

[Figure]

[Figure]

   i.flood 201808151200             j.flood 201905251600

  Figure 7 Flood simulation effects based on initial and optimized parameters.

---

## Community Comment (CC5)

**Paper Review:**

**"A physically-based distributed karst hydrological model (QMG 2 model-V1.0) for flood simulations".**

*This review was prepared as part of graduate program Earth & Environment (course Integrated Topics in Earth & Environment) at Wageningen University, and has been produced under supervision of dr Ryan Teuling. The review has been posted because of its potential usefulness to the authors and editor. Although it has the format of a regular review as was requested by the course, this review was not solicited by the journal, and should be seen as a regular comment. We leave it up to the author's and editor which points will be addressed.*

**Header**

The paper describes a modelling study of the karst trough valley Qingmuguan in China. Nowadays, due to climate change more extreme hydrological events such as flooding happen. Karst areas have very complex structure and therefore it is hard to mimic such hydrological systems. Complex models are able to model such systems but have high uncertainty. Therefore, this paper proposes a simple physically-based hydrological model (QMG) to accurately predict flooding at the Qingmuguan karst area. The modelling set up of the area is explained and the main calculations are explained. Thereafter, the model parameters are calibrated and optimized using the improved PSO algorithm IPSO. The sensitivity of each parameter was analysed. Eventually, the Nash-Sutcliffe coefficient (NSC) of the model is calculated such that the model performance can be determined. The optimization is that the model performance (NSC) increases form 0.69 to 0.92. According the authors it is concluded that this model is therefore very 'accurate' and thus suitable for predicting flooding in the karst trough valley Qingmuguan.

The novel aim of the authors is to overcome the high modelling-data demand by finding a simple physical-based distributed karst hydrological model which can 'accurately' predict flooding in the karst trough valley Qingmuguan. Since karst areas are very prone to flooding due to there low water bearing capacity, it is very important to find such an 'accurate' karst hydrological model.

After reading the paper, I think the authors did a great effort writing an understandable story line explaining the investigation. The aim of the paper is to accurately model and predict flooding in the karst trough valley of Qingmuguan using the simple QMG model, which needs lower modelling-data demand and a minimized model uncertainty. The entire paper is clearly dedicated to approach the aim. The methodology of the paper deals with model input, model calculations and algorithms, an optimization algorithm and an uncertainty analyses. All these steps together are necessary creating strong results to reach the aim. The figures are strong in substantiating the text and present the results really clear. The methodology is very well interconnected, but misses here and there some details which are necessary to reproduce the investigation. Moreover, the assessment to analyse the model performance is highly debatable. In this I review critically evaluate these missing pieces in the paper and give an advice how to resolve these flaws such that the author can make the paper ready to publish.

Citation:

- Li, J., Yuan, D., Zhang, F., Jiang, Y., Liu, J., & Ma, M. (2021). A physically-based distributed karst hydrological model (QMG model-V1. 0) for flood simulations. *Geoscientific Model Development Discussions*, 1-44. https://doi.org/10.5194/gmd-2021-120

**Major arguments**

Reading the paper and considering the quality of especially the methodology I jumped into some fundamental flaws. It is necessary to resolve these missing pieces before publishing the paper.

*Modified formulas from literature*

In the paper the authors refer in the methodology a lot to formulas used from the literature. Many of these formulas are 'modified' from literature. This is the case for formulas 1, 2, 3, 12 and 14. However, the authors do not explain what is modified about these formulas.

The authors should explain step for step what and how the formulas are deduced and modified, such that these modifications can be checked on validity.

*Darcy-Weisbach transformation*

In the methodology the authors formulate the equation used to calculate the conduit water flow of underground river to the outflow basin for turbulent conditions using Darcy-Weisbach. For this formula the authors refer to Shoemaker (2008). Li (2021) used the following formula which represents Darcy-Weisbach:

$$Q_{turbulent} = -2A\sqrt{\frac{2gd\,|\Delta h|}{\Delta l \tau}} \log\left(\frac{H_c}{3.71d} + \frac{2.51v}{d\sqrt{\frac{2gd^3\,|\Delta h|}{\Delta l \tau}}}\right)\frac{\Delta h}{|\Delta h|}$$

*Figure 1 Transformed Darcy-Weisbach formula (Li et al, 2021)*

The paper of Shoemaker only describes from which previous studies he deduced a formula for Darcy-Weisbach. Looking further at the general equation of Darcy-Weisbach, I found a different formula than used in this paper (Li et al, 2021). The formula seems to be transformed but it is not clear how they transformed this formula. Therefore, I think the authors should include a description of how they came up with the formula they used as Darcy-Weisbach. Because the way it is now described I was not able to check whether this Darcy-Weisbach equation is correctly deduced and transformed.

I would propose the authors to use the formulas given in the paper Valiantzas (2008) where the formulas used for pipe flow in a turbulent state are described. Using these fundamental equations as starting point they can show how they transformed the formula and they can confirm their transformation such that the use of their notation of Darcy-Weisbach is validated.

*Use of IPSO*

Beside describing the model environment and the main model calculations, it is of interest to describe how the authors approached to make the model as accurate as possible. Here, in this paper there is an algorithm used to calibrate and optimize the model parameters such that the uncertainty is minimized. This paper clearly describes how they started thinking about which algorithm to use for this explicit task. They came up with the algorithm PSO. However, PSO has some weaknesses and therefore they decided to improve this algorithm and they came up with the IPSO algorithm which includes random processes. A formula for this algorithm was given without any citations. Moreover, it is not entirely clear how they came up with the improvement of PSO by adding the chaotic behaviour resulting in IPSO. Therefore, I consulted literature to get acquainted of the use of ISPO algorithm (Abdi et al, 2013). Reading this paper and its formula for the IPSO algorithm, I saw that there are two main differences compared to the paper about karst modelling (Ji Li et al, 2021).

$$X_{ij}^k + 2 \times rand \times (Pbest_{ij} - X_{ij}^k) \quad \text{if } rand() < \gamma;$$

*Figure 2 IPSO algorithm equation (Abdi et al, 2013)*

$$\begin{cases} X_{ij} = X_{min} + (X_{max} - X_{min}) * Z_{ij} \\ Z'_{ij} = (1-\alpha)Z^* + \alpha Z_{ij} \end{cases}$$

*Figure 3 IPSO algorithm equation (Li et al, 2021)*

The first difference is that the paper of Li (2021) uses $x_{min}$ where the other paper uses the notation $x^k_{ki}$ and where Li (2021) uses Pbest$_{ij}$ where the other paper uses $x_{max}$. . After further reading I found out that $x_{min} = x^k_{ki}$ and that $x_{max} =$ Pbest$_{ij}$, so this first difference is only a notation difference.

The second difference is the equation used to describe the random processes. Li (2021) used an equation where an algorithm 'a', which is a variable determined by the adaptive algorithm, is used together with $Z_{ij}$, which is the variable before the disturbance is added, to describe the chaotic processes. The other paper just uses the regular formula 'rand()' to describe the chaotic processes. It is not clear in the paper of Li (2021) why they used this different function for $Z_{ij}$ to describe the random processes. It is also not clear how he came up with this formula (deduced by himself or from literature).

They should mention this in the methodology and not just introduce this formula out of nowhere. I propose to the authors to either show how they came up with the formula describing the chaotic processes substantiated by a reference or to use the same formula for the chaotic behaviour as used by Abdi (2013).

*Model performance; debatable indictor NSC*

In the paper the authors used the well known Nash-Sutcliff coefficient (NSC) to asses the model performance. However, is this NSC sufficient to assume whether a model is 'accurate'. In order to get an answer to this I consulted literature about the NSC and found a clear conclusion from this *(e.g., McCuen et al, 2006)*:

*"Many factors influence a sample value of Nash–Sutcliffe coefficient (NSC), and high values of NSC may result even when the fit is relatively poor, such as when the variance of Y is very large. Values of NSC also depend on the sample size, such that the interpretation of "good" versus "bad" fit depends on the sample size. A value of 0.7 may or may not be indicative of a good fit. Therefore, if the Nash–Sutcliffe index is to be used with some sense of reliability, more knowledge about sample values of NSC is needed".*

This makes the statement of the author unreliable. This is very important since this NSC determines the model performance and thus the accuracy of the model outcome. Thus, the statement/conclusion of the authors, that the QMG model is 'accurate' in predicting flooding in the Qingmuguan karst trough valley, questionable. This is the answers to the main research question of the investigation and therefore it is crucial that the outcome is reliable.

To resolve the shortcoming of the NSC, Schaefli & Gupta did investigation and published a paper (2007). To properly communicate how good a model really is, it seems necessary to establish appropriate reference or benchmark models; models having an easy-to-apprehend explanatory power for a given case study and a given modelling time step (e.g., Schaefli & Gupta, 2007:, e.g. Seibert, 2001). Therefore, I propose to use, instead of the mean of the observations, a benchmark model which could serve as reference. Such a benchmark model is a very simple model, applicable on the investigation, which simulates the scenario of interest in a very simplified manner. In this case, where we have in and out flow through the karst trough valley, we can use as benchmark model for example the linear reservoir model. The linear reservoir model is a simple mathematical model describing the rainfall–runoff relations of a rainfall catchment area/basin. Using this simple model as benchmark model, to obtain the NSC (instead of using the mean of the observation), we provide an analyses which should fulfil the basic requirement that every hydrologist can immediately understand
its explanatory power for the given case study and, therefore, appreciate how much better the actual hydrologic model is (e.g., Schaefli & Gupta, 2007). This would improve the quality of the analyses made in the paper. Thus, I would suggest to add this method to the model analyses.

**Minor arguments**

Minor corrections in the paper can be done as follows:

P1, line 1: The title seems to suggest that you model floods in karst area, but in the paper you propose a model explicitly for the Qingmuguan karst trough valley. This should be added in the title otherwise it is misleading.

P5, line 113: Here you state that especially the sensitivity of the parameters are analysed. Say here that this is very important to do but that this is subservient to the main aim of this research: Accurately predict flooding in the karst trough valley Qingmuguan using the simple QMG model.

P17, line 415: Add reference for the use of IPSO formula.

P18, line 448: Add reference for the use of the parametric priori distribution.

P25, line 637: Unclear what do to with the sensitivity classification of the parameters. Explain briefly the purpose of this analyses/classification and how you accounted for this considering the outcome.

P24, line 594: When discussing the lack of accuracy on predicting the dry season runoff, explain briefly the definition of this feature.

P27, line 686: I propose to mention the Nash coefficient obtained for the model performance using the initial model parameters (NSC=0.69) and the Nash coefficient obtained for the model performance after calibration and optimization of the model parameters (NSC=0.92). Now the relative improvements you mention seem new and confusing to me.

P41, line 941: Redundant figure. You can consider to remove this figure. You can easily show the KHRUs on figure 1a or 1b.

Instead of listing all figures and tables at the end of the document, it is better to put the figures in line with the text where it corresponds to. This way it makes the reading process more fluent and pleasant.

**References**

- Abdi, S., & Afshar, K. (2013). Application of IPSO-Monte Carlo for optimal distributed generation allocation and sizing. *International Journal of Electrical Power & Energy Systems*, *44*(1), 786-797.
- McCuen, R. H., Knight, Z., & Cutter, A. G. (2006). Evaluation of the Nash–Sutcliffe efficiency index. *Journal of hydrologic engineering*, *11*(6), 597-602.
- Schaefli, B., & Gupta, H. V. (2007). Do Nash values have value?. *Hydrological Processes*, *21*(ARTICLE), 2075-2080.
- Seibert J. 2001. On the need for benchmarks in hydrological modelling. Hydrological Processes 15: 1063–1064. DOI: 10.1002/ hyp.446.
- Shoemaker, W. B., Cunningham, K. J., Kuniansky, E. L., & Dixon, J. (2008). Effects of turbulence on hydraulic heads and parameter sensitivities in preferential groundwater flow layers. *Water resources research*, *44*(3).
- Valiantzas, J. D. (2008). Explicit power formula for the Darcy–Weisbach pipe flow equation: application in optimal pipeline design. *Journal of irrigation and drainage engineering*, *134*(4), 454-461.

---

## Community Comment (CC7)

**Peer Review**

**Article reviewed:** *A physically-based distributed karst hydrological model (QMG 2 model-V1.0) for flood simulations (Li et al., 2021).*

**Header**

In this paper a physically based, distributed hydrological model, (the Qingmuguan model, QMG), is developed to predict runoff and confluence for a karst trough valley. The QMG model has a simple two-layer structure: a surface part, for modelling runoff and confluence, and an underground compartment for the sub-surface river system. 18 floods recorded between 2017-2019 were used to calibrate and validate the model for the small the Qingmuguan karst valley in China. Sensitivity analysis on 10 of the 12 model parameters has indicated the following order of parameter sensitivity: infiltration coefficient > permeability coefficient > rock porosity > specific yield > saturated water content > field capacity > flow direction > thickness > slope > soil coefficient > channel roughness > evaporation. After model optimization, the Nash-Sutcliffe coefficient, correlation coefficient, water balance coefficient, average relative flow process error, flood peak error, and peak time error were 0.92, 0.90, 10%, 11%, 0.92 and 2h respectively. The study has clear novelty in that it proposes a new simplified model for flood simulations in karst areas that has a limited number of parameters, which is openly available and easily accessible. The paper has a good overall structure, the order of steps taken within the research are sound and well described. Next, different evaluation indices were used to assess the model performance and the model outcome showed to improve for all these indices after it was optimized. In short, the model gives a good overall view of water transported in the specific type of karst area described, as it links surface and sub-surface flow. In general, his paper provides a good starting point for the simulation of peak discharge in karst valleys that are largely affected by topography. The core research has been conducted properly, however, some recommendations are suggested that will help to also communicate this key message and will and put it into context, which need to be implemented before it can be published. The paper already contains the necessary buildings blocks, which individually are strong components, however the linkages between them need to be improved to capture the main findings of the study. The main issue to address is the assessment of model performance, here additional steps are required to claim the model performance is acceptable. Also, the explanatory power of the model is limited in that further elaboration and justification is needed to describe the link between the model and physical processes that occur. Lastly, the relevance of the model to the scientific community is questioned, as only its application to a very small and specific karst valley was tested. To conclude, the work can be accepted after minor revisions will be made.

**Major argument #1**

To start with the first issue on the assessment of the performance of the model, in line 43 equation 18 is given for the calculation of the Nash- Sutcliffe coefficient. The observed squared difference between the discharge minus the simulated discharge is divided by the squared simulated discharge minus the average value of observed discharge. In table 4, the Nash Sutcliffe coefficients are then given for the improved model results. In line 582 it is stated the Nash Sutcliffe values was 0.79 before parameter optimization and increased to 0.92 after optimization, which is later labelled a reasonable flood simulation result (line 588). However, based on this approach it cannot be directly claimed that the model effectively simulates peak flow in the karst valley, I think additional steps need to be added to this method. Since the Nash-Sutcliffe coefficient is computed with squared values, large discharge values as observed during peak flows are over estimated (Krause et al., 2005), which is a general remark of this method. More importantly, the Nash-Sutcliffe assessment only provides a relative indication of

the model performance, nothing can be said about performance in absolute terms, it only gives an indication about the amount of noise generated by the model compared to the signal generated. Thus, the peak flow prediction of the QMG model is better compared to the reference model, in this case the average value of observed discharges over the observation period of two years. Yet, if discharge is averaged this reduces the information on peak discharge, which is the main variable of interest. Therefore, I would recommend that the model is evaluated with a Nash-Sutcliffe efficiency that uses a well-defined benchmark model, suitable for the variable of interest, for example the calendar day benchmark model that uses an interannual average value for every calendar day, as suggested by Schaefli & Gupta (2007).

**Major argument #2**

The next issue is the relevance to the scientific community. The model aims to provide a simple model to simulate floods in karst trough valleys. In section 698 hesitance regarding its applicability to other regions is addressed. The model was calibrated and validated only for this specific karst area, which is a small basin of 13.4km2 (line 118). Many other karst areas occur in China, let alone globally, which would vouch this model's relevance. However, as noted in this study and many others, for example Bakalowicz (2005), modelling water flow in karst areas is very complex due to the heterogeneity of site characteristics such as the extent of network of conduits. Consequently, the QMG model might not generate accurate results for larger areas, due to the spatial variety present in such areas. Therefore, I would like to recommend researching additional methods that can be used to guarantee effective simulation of flow through larger areas, that add to the tracer test method that was already mentioned.

**Major argument #3**

The next issue is that it is not well supported whether the model represents reality accurately, the performance of the model is not put into context. In line 685 it is stated that the flood simulation after parameter optimization with IPSO was much better than simulation of the initial model parameters, and that the six indicators of model performance demonstrated increase overall outcome of the model after optimization (line 687-698). However, this indicator only shows that the optimized parameters score better compared to the initial parameters, the actual physical processes are not directly involved in this. The gap between the conceptual model constructed and the actual physical processes is unaddressed, so these optimized parameters are no guarantee for exact representation of the physical reality. This has consequences in the sensitivity analysis, which is the next part of the study. In line 694 it is concluded that the rainfall infiltration coefficient is the most sensitive parameter. However, this is based on the representation of reality that is constructed in the model. Therefore, I would recommend to firstly add a sketch of the conceptual model in the methodology section, after the separate equations for the main processes are discussed. Secondly, I would add additional sources that justify this conceptualization of the system, so previous studies that adopted a similar approach, for example Epting et al. (2018).

**Minor arguments**

- In line 451 a Nash-coefficient of 0.85 is mentioned as threshold, above which the model performance is labelled acceptable, although this number intuitively makes sense it needs be explained or supported by a reference.
- To add to the assessment of the model performance for dry period simulation the log-transformation of the Nash-Sutcliffe coefficient can be calculated.
- It is not always explicitly stated what is meant inexplicitly. For example, the aim and RQ are included in the first section of the paper, and from reading the overall paragraph you can deduce the aim/RQ yourself, but it is not directly stated, so this needed to be stated more clearly.
- In equation 2, the parameter $f_i$ is not explained in the text below

- It is not clear why 10 cycles were chosen for the IPSO parameter optimization (line 412), a justification for this number should be added to, why does 10 cycles result in global convergence of the model?
- For figure 1a, the legend is quite difficult to identify, especially icon 4 and 5 are very hard to find in the map → increase size of those figures in the map
- For figure 6 the dates displayed on the x-axis are very hard to read. I would suggest displaying less dates (for example only months). Next the label of the simulated Q, both with IPSO and the initial parameters are misleading; it seems as if the *simulated Q* is *divided* by the *IPSO* and *Initial parameters*. This same confusion of labels is present in the graphs of figure 7.

**Typos**

The language needs some extra revision, below a list of example typos is given that should be improved:

- Line 19: strcutrues → structures
- Line 39: the "." before "Because" should be a comma
- Line 44: add a comma between "increased" and "that", the sentence is really long now
- Line 94: "which make distributed model may need": there is an error in this sentence, it is not readable
- Line 108: "work" → add 's'
- Line 109: between parameters and we, the dot needs to be a comma
- Line 177: "Berry et" → 'al.' misses from the reference
- Line 675: after "follow", change the dot to a colon

**References**

Bakalowicz, M. (2005). Karst groundwater: a challenge for new resources. Hydrogeology journal, 13(1), 148-160.

Epting, J., Page, R. M., Auckenthaler, A., & Huggenberger, P. (2018). Process-based monitoring and modeling of Karst springs–linking intrinsic to specific vulnerability. Science of the total environment, 625, 403-415.

Krause, P., Boyle, D. P., & Bäse, F. (2005). Comparison of different efficiency criteria for hydrological model assessment. Advances in geosciences, 5, 89-97.

Schaefli, B., & Gupta, H. V. (2007). Do Nash values have value? Hydrological Processes, 21(ARTICLE), 2075-2080.

---

## Author Response (AR1)

**A point-by-point reply to the comments**

Anonymous Referee #2, 19 Feb 2022

The paper deals with implementation of a physically-based distributed karst hydrological model for flood simulations. The manuscript has several deficiencies, in part depending upon the English language and in part by problems with the scientific content.

I have a number of considerations and suggestions (presented in this comment, and in the attached file as well).

In general, the English language seems to me not satisfying the international standards for publication in several points, and needs some deep revision. In particular, I pointed out in the attached file some parts where the English was unclear to me.

Authors are probably not very familiar with karst literature and terminology. Since they are proposing a model for floods in karst, the karst literature cannot be not taken into account. From the beginning, it is stated that the works regards "karst though valley". This is not a term familiar to me, and I do not have seen it used in the karst literature. Thus, its meaning should be clearly defined. In addition, references to the main works and textbooks as concerns karst landforms and morphology should be added. Below you will find some suggestions at this regard.

As regards the main topic of the article, that is floods, karst settings are typically characterized by flash floods, due to the lack or scarcity of water at the surface during most of the year. This is never mentioned in the manuscript, but should deserve some mention, also to cite similar examples in other karst areas worldwide. For instance, have a look at the paper by Gutierrez et al. (2014) and the abundant references about floods in karst (Parise, 2003; Bonacci et al., 2006; Jourde et al., 2007, 2014; Martinotti et al., 2017).

Also when dealing with sinkholes, no reference to the main classification of sinkholes is provided. All this indicate a quite poor knowledge of karst, which should be addresses for an article submitted to international journals.

Suggested references for karst (general textbooks and specific articles for floods and hazards in karst):

Bonacci, O., Ljubenkov, I., Roje-Bonacci, T., 2006. Karst flash floods: an example from the Dinaric karst Croatia. Nat. Hazards Earth Syst. Sci. 6, 195–203.

Ford, D.C.,Williams, P., 2007. Karst Hydrogeology and Geomorphology.Wiley, Chichester, 562 pp..

Gutierrez, F., 2010. Hazards associated with karst. In: Alcantara, I. & A. Goudie (Eds.), Geomorphological Hazards and Disaster Prevention. Cambridge University Press, Cambridge, 161–175.

Gutierrez F., Parise M., De Waele J. & Jourde H., 2014, A review on natural and human-induced geohazards and impacts in karst. Earth Science Reviews, vol. 138, p. 61-88.

Jourde, H., Roesch, A., Guinot, V., Bailly-Comte, V., 2007. Dynamics and contribution of karst groundwater to surface flow during Mediterranean flood. Environ. Geol. 51 (5), 725–730.

Jourde, H., Lafare, A., Mazzilli, N., Belaud, G., Neppel, L., Doerfliger, N., Cernesson, F., 2014. Flash flood mitigation as a positive consequence of anthropogenic forcings on the groundwater resource in a karst catchment. Environ. Earth Sci. 71, 573–583.

Martinotti M.E., Pisano L., Marchesini I., Rossi M., Peruccacci S., Brunetti M.T., Melillo M., Amoruso G., Loiacono P., Vennari C., Vessia G., Trabace M., Parise M., & Guzzetti F., 2017, Landslides, floods and sinkholes in a karst environment: the 1–6 September 2014 Gargano event, southern Italy. Natural Hazards and Earth System Sciences, vol. 17, p. 467-480.

Palmer, A.N., 2007. Cave Geology. Cave Books, Dayton, 454 pp..

Palmer, A.N., 2010. Understanding the hydrology of karst. Geol. Croat. 63, 143–148.

Parise, M., 2003. Flood history in the karst environment of Castellana-Grotte (Apulia, southern Italy). Nat. Hazards Earth Syst. Sci. 3 (6), 593–604.

Parise, M., 2010, Hazards in karst, Proceedings Int. Conf. "Sustainability of the karst environment. Dinaric karst and other karst regions", IHP-Unesco, Series on Groundwater, 2, 155-162.

Parise M., Ravbar N., Å½ivanovic V., Mikszewski A., Kresic N., Mádl-Szo Ì‹nyi J. & Kukuric N., 2015, Hazards in Karst and Managing Water Resources Quality. Chapter 17 in: Z. Stevanovic (ed.), Karst Aquifers – Characterization and Engineering. Professional Practice in Earth Sciences, Springer, pp. 601-687.

White, W.B., 1988. Geomorphology and Hydrology of Karst Terrains. Oxford University Press, Oxford, 464 pp.

White, W.B., 2002. Karst hydrology: recent developments and open questions. Eng. Geol. 65, 85–105.

Williams, P.W., 2008. The role of the epikarst in karst and cave hydrogeology: a review. Int. J. Speleol. 37, 1–10.

Zhou W, Beck BF (2011) Engineering issues on karst. In: P. van Beynen (Ed), Karst Management. Springer, Dordrecht, 9-45.

Please also note the supplement to this comment:

https://gmd.copernicus.org/preprints/gmd-2021-120/gmd-2021-120-RC2-supplement.pdf

**General comment:**

The paper deals with implementation of a physically-based distributed karst hydrological model for flood simulations. The manuscript has several deficiencies, in part depending upon the English language and in part by problems with the scientific content.

I have a number of considerations and suggestions (presented in this comment, and in the attached file as well).

**Response:**

We greatly appreciate the reviewer's comments. The reviewer point out that there are two major deficiencies in this manuscript, in part depending upon the English language errors and in part by the scientific content problems. We have carefully revised the problems pointed out by the reviewer. The following is our point-by-point response to specific comments.

**Specific Comment**

1) English language errors

In general, the English language seems to me not satisfying the international standards for publication in several points, and needs some deep revision. In particular, I pointed out in the attached file some parts where the English was unclear to me.

**Response:**

We apologize for the poor  English language of the text,  and will be happy to edit the manuscript further based on the helpful comments. In fact,  the first reviewer also pointed out that the English language problems of this manuscript need to be deeply revised. Therefore, we have carefully revised the language errors in the full text, including wrong words,

terminologies, grammar and unclear sentences, etc. And to correct the language problems, we turned to a professional English editing company (Charlesworth Advanced ) to help fix the language errors in the manuscript.

2) Terminology problems

Authors are probably not very familiar with karst literature and terminology. Since they are proposing a model for floods in karst, the karst literature cannot be not taken into account. From the beginning, it is stated that the works regards "karst though valley". This is not a term familiar to me, and I do not have seen it used in the karst literature. Thus, its meaning should be clearly defined.

**Response:**

As mentioned above, we have corrected the terminology errors in the text, for instance, the karst though valley was replaced by karst though and valley landform (Lines 37); skylights was replaced by karst windows (Lines 311).

3) References

In addition, references to the main works and textbooks as concerns karst landforms and morphology should be added. Below you will find some suggestions at this regard.

**Response:**

The reviewer pointed out literature citations are inadequate, and references to karst landforms and morphology should be added. Thus, we have added some relevant literatures based on the suggestions (most of them were added in the Introduction part, for example, Lines 41, 48-50, and Lines 54, 58, etc.).

As regards the main topic of the article, that is floods, karst settings are typically characterized by flash floods, due to the lack or scarcity of water at the surface during most of the year. This is never mentioned in the manuscript, but should deserve some mention, also to cite similar examples in other karst areas worldwide. For instance, have a look at the paper by Gutierrez et al. (2014) and the abundant references about floods in karst (Parise, 2003; Bonacci et al., 2006; Jourde et al., 2007, 2014; Martinotti et al., 2017).

**Response:**

The coexistence of drought and flood is a typical phenomenon in karst areas, and Indeed, as the reviewer stated, the water shortage in karst areas for most of the year. Therefore, the lack or scarcity of water problems have been added in the text (Lines 38-39 and 41-42), and some relevant literatures have also been added accordingly (Lines 48-50).

Also when dealing with sinkholes, no reference to the main classification of sinkholes is provided. All this indicate a quite poor knowledge of karst, which should be addresses for an article submitted to international journals.

**Response:**

References to the main classification of karst sinkholes have been added in the revised version (Lines 163-164, and Lines 202-207).

Suggested references for karst (general textbooks and specific articles for floods and hazards in karst):

Bonacci, O., Ljubenkov, I., Roje-Bonacci, T., 2006. Karst flash floods: an example from the Dinaric karst Croatia. Nat. Hazards Earth Syst. Sci. 6, 195–203.

Ford, D.C.,Williams, P., 2007. Karst Hydrogeology and Geomorphology.Wiley, Chichester, 562 pp..

Gutierrez, F., 2010. Hazards associated with karst. In: Alcantara, I. & A. Goudie (Eds.), Geomorphological Hazards and Disaster Prevention. Cambridge University Press, Cambridge, 161–175.

Gutierrez F., Parise M., De Waele J. & Jourde H., 2014, A review on natural and human-induced geohazards and impacts in karst. Earth Science Reviews, vol. 138, p. 61-88.

Jourde, H., Roesch, A., Guinot, V., Bailly-Comte, V., 2007. Dynamics and contribution of karst groundwater to surface flow during Mediterranean flood. Environ. Geol. 51 (5), 725–730.

Jourde, H., Lafare, A., Mazzilli, N., Belaud, G., Neppel, L., Doerfliger, N., Cernesson, F., 2014. Flash flood mitigation as a positive consequence of anthropogenic forcings on the groundwater resource in a karst catchment. Environ. Earth Sci. 71, 573–583.

Martinotti M.E., Pisano L., Marchesini I., Rossi M., Peruccacci S., Brunetti M.T., Melillo M., Amoruso G., Loiacono P., Vennari C., Vessia G., Trabace M., Parise M., & Guzzetti F., 2017, Landslides, floods and sinkholes in a karst environment: the 1–6 September 2014 Gargano event, southern Italy. Natural Hazards and Earth System Sciences, vol. 17, p.

467-480.

Palmer, A.N., 2007. Cave Geology. Cave Books, Dayton, 454 pp..

Palmer, A.N., 2010. Understanding the hydrology of karst. Geol. Croat. 63, 143–148.

Parise, M., 2003. Flood history in the karst environment of Castellana-Grotte (Apulia, southern Italy). Nat. Hazards Earth Syst. Sci. 3 (6), 593–604.

Parise, M., 2010, Hazards in karst, Proceedings Int. Conf. "Sustainability of the karst environment. Dinaric karst and other karst regions", IHP-Unesco, Series on Groundwater, 2, 155-162.

Parise M., Ravbar N., Å½ivanovic V., Mikszewski A., Kresic N., Mádl-Szo Ì‹nyi J. & Kukuric N., 2015, Hazards in Karst and Managing Water Resources Quality. Chapter 17 in: Z. Stevanovic (ed.), Karst Aquifers – Characterization and Engineering. Professional Practice in Earth Sciences, Springer, pp. 601-687.

White, W.B., 1988. Geomorphology and Hydrology of Karst Terrains. Oxford University Press, Oxford, 464 pp.

White, W.B., 2002. Karst hydrology: recent developments and open questions. Eng. Geol. 65, 85–105.

Williams, P.W., 2008. The role of the epikarst in karst and cave hydrogeology: a review. Int. J. Speleol. 37, 1–10.

Zhou W, Beck BF (2011) Engineering issues on karst. In: P. van Beynen (Ed), Karst Management. Springer, Dordrecht, 9-45.

**Response:**

Some of the above mentioned references for karst have been added to the article (for example, References: Lines 813-814, 861-865,876-881,and 932-939, etc, and the corresponding literatures have also been added to the text).

Please also note the supplement to this comment:

https://gmd.copernicus.org/preprints/gmd-2021-120/gmd-2021-120-RC2-supplement.pdf

**Response:**

We have carefully revised the errors based on the comments in the supplement file (gmd-2021-120-RC2-supplement.pdf), and these changes are highlighted in revision mode (a marked-up manuscript.docx).

---

## Author Response (AR2)

**Point-by-point replies to the comments**

**1. Topical Editor decision: Reconsider after major revisions**

by Bethanna Jackson 28 Feb 2022

Comments to the author:

Dear authors, I think this paper is significantly improved and I do think we can get it over the line to just technical or minor revisions if not immediate acceptance with one more iteration if you are willing to put an effort into this further iteration in. When I first accepted it for review, I thought there were both many strengths but also some significant weaknesses and most of these latter are now addressed. I want to acknowledge to you I feel the process of taking it to publication is more complex than usual, not through any faults of yours. Not just because of some translation issues into English, but also because karst as you know is complex and geospatially variable. You have had some very competent and constructive reviews from experts in European and other global karst landscapes and I suspect (partially from reading your descriptions, and also from one trip to China where I viewed karst landscapes among others) that you have differences in the overall qualities and responses of the karst landscapes versus those published in the English literature. It's great you have brought in the English literature, and please don't hesitate to add into the paper differences between karst descriptions and behaviours in that published literature and what you are trying to model- that may really help later authors to reference descriptions of Chinese/other less published specific karstic landscapes. Very best wishes, Beth

**2. Anonymous Reviewer #1, 20 Sep 2021**

The paper concerns a topic consistent with the aim of the GMD journal, and I really appreciate the huge work made by the authors. The presented analysis and model application could be potentially useful in karst basins. In this study, a karst hydrological model, i.e., the QMG model-V1.0 was developed for karst floods simulation and forecasting. The model itself is a valuable improvement, and what interested me was the applicability of the model in karst areas, so I went through the entire process of modeling and validating the model myself (https://zenodo.org/deposit?page=1&size=20), and the model simulation results were satisfactory. I think the subsequent research should focus on the validation study of the model in more karst areas to prove its general applicability in karst hydrological forecasting. However, there are few drawbacks affect the manuscript and have to be addressed before the paper can be published in GMD.

Specific comments

1) English needs modification

I found several incorrect words, grammar and unclear sentences, make it very difficult to

understand the analysis carried out and the results obtained. The authors need to carefully correct the language errors in the whole text.

2) More information about the potential of this new model, ie.e., the QMG model-V1.0 for application in karst areas needs to be added in the Introduction part, especially the advantages and disadvantages compared to current numerical karst groundwater models.

3) In the Methodology part, the section 3.1 Hydrological model, this title is inappropriate here, as it obviously also includes the Parameter Optimization in Section 3.2 and Model Setting in 3.4. Suggest changing it to a model framework and algorithm.

4) In section 3.3 Uncertainty Analysis, it is not clear how to analyze uncertainty in input data and model structure for this new QMG model-V1.0.

Other minor comments

1) All tables should be set to three-line tables.

2) The right side of Figure 3 seems to be a photograph, please explain the necessity of its existence.

3) Each variable in Figure 5 needs to be clearly labeled as to which parameter it refers to.

4) The horizontal axis in Figure 7 represents the date, but the interval is not one-to-one with the marked time, please check that.

**Anonymous Reviewer #2, 19 Feb 2022**

The paper deals with implementation of a physically-based distributed karst hydrological

model for flood simulations. The manuscript has several deficiencies, in part depending

upon the English language and in part by problems with the scientific content.

I have a number of considerations and suggestions (presented in this comment, and in the

attached file as well).

In general, the English language seems to me not satisfying the international standards

for publication in several points, and needs some deep revision. In particular, I pointed

out in the attached file some parts where the English was unclear to me.

Authors are probably not very familiar with karst literature and terminology. Since they

are proposing a model for floods in karst, the karst literature cannot be not taken into

account. From the beginning, it is stated that the works regards "karst though valley". This is not a term familiar to me, and I do not have seen it used in the karst literature. Thus, its meaning should be clearly defined. In addition, references to the main works and textbooks as concerns karst landforms and morphology should be added. Below you will find some suggestions at this regard.

As regards the main topic of the article, that is floods, karst settings are typically characterized by flash floods, due to the lack or scarcity of water at the surface during most of the year. This is never mentioned in the manuscript, but should deserve some mention, also to cite similar examples in other karst areas worldwide. For instance, have a look at the paper by Gutierrez et al. (2014) and the abundant references about floods in karst (Parise, 2003; Bonacci et al., 2006; Jourde et al., 2007, 2014; Martinotti et al., 2017).

Also when dealing with sinkholes, no reference to the main classification of sinkholes is provided. All this indicate a quite poor knowledge of karst, which should be addresses for an article submitted to international journals.

Suggested references for karst (general textbooks and specific articles for floods and hazards in karst):

Bonacci, O., Ljubenkov, I., Roje-Bonacci, T., 2006. Karst flash floods: an example from the Dinaric karst Croatia. Nat. Hazards Earth Syst. Sci. 6, 195–203.

Ford, D.C.,Williams, P., 2007. Karst Hydrogeology and Geomorphology.Wiley, Chichester, 562 pp..

Gutierrez, F., 2010. Hazards associated with karst. In: Alcantara, I. & A. Goudie (Eds.), Geomorphological Hazards and Disaster Prevention. Cambridge University Press, Cambridge, 161–175.

Gutierrez F., Parise M., De Waele J. & Jourde H., 2014, A review on natural and human-induced geohazards and impacts in karst. Earth Science Reviews, vol. 138, p. 61-88.

Jourde, H., Roesch, A., Guinot, V., Bailly-Comte, V., 2007. Dynamics and contribution of karst groundwater to surface flow during Mediterranean flood. Environ. Geol. 51 (5), 725–730.

Jourde, H., Lafare, A., Mazzilli, N., Belaud, G., Neppel, L., Doerfliger, N., Cernesson, F., 2014. Flash flood mitigation as a positive consequence of anthropogenic forcings on the groundwater resource in a karst catchment. Environ. Earth Sci. 71, 573–583.

Martinotti M.E., Pisano L., Marchesini I., Rossi M., Peruccacci S., Brunetti M.T., Melillo M., Amoruso G., Loiacono P., Vennari C., Vessia G., Trabace M., Parise M., & Guzzetti F., 2017, Landslides, floods and sinkholes in a karst environment: the 1–6 September 2014 Gargano event, southern Italy. Natural Hazards and Earth System Sciences, vol. 17, p. 467-480.

Palmer, A.N., 2007. Cave Geology. Cave Books, Dayton, 454 pp..

Palmer, A.N., 2010. Understanding the hydrology of karst. Geol. Croat. 63, 143–148.

Parise, M., 2003. Flood history in the karst environment of Castellana-Grotte (Apulia, southern Italy). Nat. Hazards Earth Syst. Sci. 3 (6), 593–604.

Parise, M., 2010, Hazards in karst, Proceedings Int. Conf. "Sustainability of the karst environment. Dinaric karst and other karst regions", IHP-Unesco, Series on Groundwater, 2, 155-162.

Parise M., Ravbar N., Å½ivanovic V., Mikszewski A., Kresic N., Mádl-Szo Ì‹nyi J. & Kukuric N., 2015, Hazards in Karst and Managing Water Resources Quality. Chapter 17 in: Z. Stevanovic (ed.), Karst Aquifers – Characterization and Engineering. Professional Practice in Earth Sciences, Springer, pp. 601-687.

White, W.B., 1988. Geomorphology and Hydrology of Karst Terrains. Oxford University Press, Oxford, 464 pp.

White, W.B., 2002. Karst hydrology: recent developments and open questions. Eng. Geol. 65, 85–105.

Williams, P.W., 2008. The role of the epikarst in karst and cave hydrogeology: a review.

Int. J. Speleol. 37, 1–10.

Zhou W, Beck BF (2011) Engineering issues on karst. In: P. van Beynen (Ed), Karst

Management. Springer, Dordrecht, 9-45.

Please also note the supplement to this comment:

https://gmd.copernicus.org/preprints/gmd-2021-120/gmd-2021-120-RC2-supplement.pdf

**Response to Topical Editor' comments**

First, many thanks to the Topical Editor for acknowledging that this paper has been significantly improved. We are willing to revise and improve this paper based on the comments of the reviewers and Editor. Indeed, as stated by the Editor-in-Chief, karst geography is complex and variable. There are differences in the overall quality and response of the karst landscape versus those published in the literature.

In the revised manuscript, we have added the differences between karst descriptions and behaviours in the published literature and those in our QMG model in this karst study area. For instance, the karst trough landform in this area looks like a pen-holder structure, means 'three ridges with two troughs' (Lines 149-151). which provides convenient conditions for flood propagation and formation. Such trough and valley landforms may not be common in other karst regions of the world, but they are common and typical in the karst regions of southwest China, especially Chongqing. In addition, both reviewers pointed out that the English language in the paper requires substantial improvement to enhance readability. In particular, the second reviewer highlighted the irregular use of karst terminology in the manuscript, which resulted in the article appearing unprofessional. Therefore, we have carefully revised the English language errors in this paper, especially the karst terminology, and asked a professional English editing company (Charlesworth Advanced ) to help fix the language problems in the text so that readers can better understand the ideas and scientific conclusions expressed in this manuscript.

**Response to the comments of Anonymous Reviewer #1**

**General comment:**

The paper concerns a topic consistent with the aim of the GMD journal, and I really appreciate the huge work made by the authors. The presented analysis and model application could be potentially useful in karst basins. In this study, a karst hydrological model, i.e., the QMG model-V1.0 was developed for karst floods simulation and forecasting. The model itself is a valuable improvement, and what interested me was the applicability of the model in karst areas, so I went through the entire process of modeling and validating the model myself (https://zenodo.org/deposit?page=1&size=20), and the model simulation results were satisfactory. I think the subsequent research should focus on the validation study of the model in more karst areas to prove its general applicability in karst hydrological forecasting. However, there are few drawbacks affect the manuscript and have to be addressed before the paper can be published in GMD.

**Response:**

We greatly appreciate the reviewer's comments. The reviewer confirmed the innovation and application value of this study, noted the potential of the model (QMG model-V1.0) proposed in karst areas and suggested that subsequent studies should focus on applying this new model to more karst areas to test its general applicability in karst flood forecasting.

The next step in our research is indeed focused on model validation, for which we will build this model (QMG model-V1.0) for flood simulation and forecasting in more karst areas and improve the model's functions and algorithms to enhance its applicability and accuracy.

Point-by-point responses to your specific comments are given below.

**Specific Comment**

1) English needs modification

I found several incorrect words, grammar and unclear sentences, make it very difficult to understand the analysis carried out and the results obtained. The authors need to carefully correct the language errors in the whole text.

**Response:**

We have carefully revised the language errors in the full text, including incorrect words, grammar errors and unclear sentences, and asked a professional English editing company (Charlesworth Advanced ) to help fix the language problems in the manuscript.

2) More information about the potential of this new model, ie.e., the QMG model-V1.0 for application in karst areas needs to be added in the Introduction part, especially the advantages and disadvantages compared to current numerical karst groundwater models.

**Response:**

More information about the advantages of the QMG model-V1.0 compared with other karst groundwater models has been added to the revised Introduction (Lines 118-130).

3) In the Methodology part, the section 3.1 Hydrological model, this title is inappropriate here, as it obviously also includes the Parameter Optimization in Section 3.2 and Model Setting in 3.4. Suggest changing it to a model framework and algorithm.

**Response:**

This advice is very valuable. The title of section 3.1 has been replaced with "Hydrological model framework and algorithms" accordingly (Lines 239).

4) In section 3.3 Uncertainty Analysis, it is not clear how to analyze uncertainty in input data and model structure for this new QMG model-V1.0.

**Response:**

Uncertainty analyses of the input data and model structure have been added to the revised section 3.3 (Lines 491-506).

Other minor comments

1) All tables should be set to three-line tables.

**Response:**

The tables have been converted to three-line tables accordingly (Lines 1071-1074).

2) The right side of Figure 3 seems to be a photograph, please explain the necessity of its existence.

**Response:**

The image shows a three-dimensional spatial model of KHRUs established in the laboratory to visually reflect the storage and movement of water in a karst water-bearing medium with spatial anisotropy and to provide technical support for the establishment of a hydrological model. This description has been added to the revised version of the paper (Lines 265-268).

3) Each variable in Figure 5 needs to be clearly labeled as to which parameter it refers to.

**Response:**

The model parameter denoted by each variable in Figure 5 is clearly listed in Table 1 (Lines 1099).

4) The horizontal axis in Figure 7 represents the date, but the interval is not one-to-one with the marked time, please check that.

**Response:**

The horizontal axis in Figure 7 has been revised accordingly (Lines 1103-1119).

**Response to the comments of Anonymous Reviewer #2**

The paper deals with implementation of a physically-based distributed karst hydrological model for flood simulations. The manuscript has several deficiencies, in part depending upon the English language and in part by problems with the scientific content.

I have a number of considerations and suggestions (presented in this comment, and in the attached file as well).

**Response:**

We greatly appreciate the reviewer's comments. The reviewer noted that there are two major deficiencies in this manuscript related to the English language errors and the scientific content problems. We have carefully revised the problems highlighted by the reviewer. Point-by-point responses to your specific comments are given below.

**Specific Comment**

1) English language errors

In general, the English language seems to me not satisfying the international standards for publication in several points, and needs some deep revision. In particular, I pointed out in the attached file some parts where the English was unclear to me.

**Response:**

We apologize for the poor English language in the text and have edited the manuscript further based on your helpful comments. The first reviewer also suggested that the English language problems in the manuscript be revised. Therefore, we have carefully addressed the language errors in the full text, including incorrect words, terminology, grammar errors and unclear sentences. We used a professional English editing company (Charlesworth Advanced ) to

help fix the language errors in the manuscript.

2) Terminology problems

Authors are probably not very familiar with karst literature and terminology. Since they are proposing a model for floods in karst, the karst literature cannot be not taken into account. From the beginning, it is stated that the works regards "karst though valley". This is not a term familiar to me, and I do not have seen it used in the karst literature. Thus, its meaning should be clearly defined.

**Response:**

As mentioned above, we have corrected the terminology errors in the text; for instance, "karst trough valley" was replaced with "karst trough and valley landform" (Lines 42), and "skylights" was replaced with "karst windows" (Lines 322).

3) References

In addition, references to the main works and textbooks as concerns karst landforms and morphology should be added. Below you will find some suggestions at this regard.

**Response:**

The reviewer noted that the literature citations are inadequate, and references to karst landforms and morphology should be added. Thus, we have added some relevant citations based on your suggestions (most of them were added in the Introduction, for example, on Lines 46, 54-55, 59, and Lines 63-64).

As regards the main topic of the article, that is floods, karst settings are typically characterized by flash floods, due to the lack or scarcity of water at the surface during most of the year. This is never mentioned in the manuscript, but should deserve some mention, also to cite similar examples in other karst areas worldwide. For instance, have a look at the paper by Gutierrez et al. (2014) and the abundant references about floods in karst (Parise, 2003; Bonacci et al., 2006; Jourde et al., 2007, 2014; Martinotti et al., 2017).

**Response:**

The coexistence of drought and flooding is a typical phenomenon in karst areas, and indeed, as the reviewer stated, water shortages in karst areas occur most of the year. Therefore, water scarcity problems are added in the text (Lines 43-44 and 46-48), and some relevant literature has been added accordingly (Lines 46).

Also when dealing with sinkholes, no reference to the main classification of sinkholes is provided. All this indicate a quite poor knowledge of karst, which should be addresses for an article submitted to international journals.

**Response:**

References related to the classification of karst sinkholes have been added in the revised version of the paper (Lines 174 and Lines 214-220).

Suggested references for karst (general textbooks and specific articles for floods and hazards in karst):

Bonacci, O., Ljubenkov, I., Roje-Bonacci, T., 2006. Karst flash floods: an example from the Dinaric karst Croatia. Nat. Hazards Earth Syst. Sci. 6, 195–203.

Ford, D.C.,Williams, P., 2007. Karst Hydrogeology and Geomorphology.Wiley, Chichester, 562 pp..

Gutierrez, F., 2010. Hazards associated with karst. In: Alcantara, I. & A. Goudie (Eds.), Geomorphological Hazards and Disaster Prevention. Cambridge University Press, Cambridge, 161–175.

Gutierrez F., Parise M., De Waele J. & Jourde H., 2014, A review on natural and human-induced geohazards and impacts in karst. Earth Science Reviews, vol. 138, p. 61-88.

Jourde, H., Roesch, A., Guinot, V., Bailly-Comte, V., 2007. Dynamics and contribution of karst groundwater to surface flow during Mediterranean flood. Environ. Geol. 51 (5), 725–730.

Jourde, H., Lafare, A., Mazzilli, N., Belaud, G., Neppel, L., Doerfliger, N., Cernesson, F., 2014. Flash flood mitigation as a positive consequence of anthropogenic forcings on the

groundwater resource in a karst catchment. Environ. Earth Sci. 71, 573–583.

Martinotti M.E., Pisano L., Marchesini I., Rossi M., Peruccacci S., Brunetti M.T., Melillo M., Amoruso G., Loiacono P., Vennari C., Vessia G., Trabace M., Parise M., & Guzzetti F., 2017, Landslides, floods and sinkholes in a karst environment: the 1–6 September 2014 Gargano event, southern Italy. Natural Hazards and Earth System Sciences, vol. 17, p. 467-480.

Palmer, A.N., 2007. Cave Geology. Cave Books, Dayton, 454 pp..

Palmer, A.N., 2010. Understanding the hydrology of karst. Geol. Croat. 63, 143–148.

Parise, M., 2003. Flood history in the karst environment of Castellana-Grotte (Apulia, southern Italy). Nat. Hazards Earth Syst. Sci. 3 (6), 593–604.

Parise, M., 2010, Hazards in karst, Proceedings Int. Conf. "Sustainability of the karst environment. Dinaric karst and other karst regions", IHP-Unesco, Series on Groundwater, 2, 155-162.

Parise M., Ravbar N., Å½ivanovic V., Mikszewski A., Kresic N., Mádl-Szo Ì‹nyi J. & Kukuric N., 2015, Hazards in Karst and Managing Water Resources Quality. Chapter 17 in: Z. Stevanovic (ed.), Karst Aquifers – Characterization and Engineering. Professional Practice in Earth Sciences, Springer, pp. 601-687.

White, W.B., 1988. Geomorphology and Hydrology of Karst Terrains. Oxford University Press, Oxford, 464 pp.

White, W.B., 2002. Karst hydrology: recent developments and open questions. Eng. Geol. 65, 85–105.

Williams, P.W., 2008. The role of the epikarst in karst and cave hydrogeology: a review. Int. J. Speleol. 37, 1–10.

Zhou W, Beck BF (2011) Engineering issues on karst. In: P. van Beynen (Ed), Karst Management. Springer, Dordrecht, 9-45.

**Response:**

Some of the abovementioned references for karst have been added to the article (for example, the references for Lines 883-884, 931-935, 946-951, and 1003-1010, etc.), and the corresponding literature has also been added to the text.

Please also note the supplement to this comment:

https://gmd.copernicus.org/preprints/gmd-2021-120/gmd-2021-120-RC2-supplement.pdf

**Response:**

We have carefully revised the errors based on the comments in the supplemental file (gmd-2021-120-RC2-supplement.pdf), and these changes have been highlighted in revision mode (a marked-up manuscript.docx).

---

## Author Response (AR3)

I have reviewed this paper once before, and the authors have satisfactorily responded to and addressed the issues I mentioned, and I think the current version of the manuscript is greatly improved and worthy of publication in this journal. Compared with the current commonly used karst groundwater models, this QMG model has the advantage of relatively simple double-layer structures and parameters, making it possible to model with a smaller data requirement in karst areas. For instance, the MODFLOW-CFP needs more distribution data of the karst underground conduit system. This QMG model can be extended to more karst areas to verify its general applicability in karst flood simulation and prediction. The innovation of this paper is clear, i.e., it proposes a hydrological model- QMG model-V1.0 applicable to karst areas, and the optimization method of model parameters-the improved particle swarm algorithm (IPSO) offered a better flood simulation effects than those with the initial model parameters, indicating the applicability of this IPSO method. In addition, the theory research on hydrological model also applies to the scope of the GMD journal. All in all, I think this paper is suitable for publication in this journal, but some minor errors need to be fixed beforehand.

**Response**

We thank you for reviewing this paper again and for your recognition of the progress of the paper and suggestions for publication. In addition, your affirmation of the innovation of this research deeply appreciated. Below are our point-by-point responses to specific comments. Specific comments

**1. Technical term**

Some terminologies in the text may be inappropriate or non-standard that need to be corrected. For example, the flood 'forecasting' had better be replaced by 'prediction';

"Excess water overflows" should be "excess infiltration runoff"; and "during floods" should be "during flooding", etc.

**Response**

The terminology in the manuscript has been effectively modified. For instance, 'forecasting' has been replaced by 'prediction' (Lines 36, etc.); "excess water overflows" has been replaced

by "excess infiltration runoff" (Line 49); and "during floods" has been replaced by "during flooding" (Lines 50).

2. The specific location of the study area in China should be indicated in Figure 1, and more information on karst distribution and development should be added to the study area and data.

**Response**

The specific location of the study area in Chongqing, China, has been added to Fig. 1.

Some information on karst distribution and development has been added to the study area and data (Lines 147-159).

3. Methodology: "3.1 Hydrological model framework and algorithms" has better be replaced by "3.1 Hydrological model"; "4.1 Parameter Sensitivity Results" should be "4.1 Parameter sensitivity results"; "4.2 Parametric Optimization" should be "4.2 Parametric optimization"; and "4.3 Model Validation in Flood Simulations" should be "4.3 Model validation".

**Response**

These headings have been modified accordingly (Lines 234, 542, 575, and 633).

4. Results and discussion should be written separately.

**Response**

Separate Results and Discussion sections have been written (Lines 541 and 662).

5. Conclusions: it is suggested to list the specific conclusions of this study one by one, so that readers can intuitively discover the main conclusions of this paper.

**Response**

The specific conclusions of this study have been listed (Lines 766, 775, and 782).

**Anonymous Referee #2**

**(1) Language errors**

Some English language errors, such as the wrong words and grammar in the text, making it a bit difficult to read and understand. It is suggested that the authors correct the language and writing problems carefully.

**Response**

The English language errors, including incorrect words and grammar, in the paper have been corrected, and a professional editing service (American Journal Experts) has addressed the language issues in the text.

(2) Study area and data

"2.3 data" should be replaced by "Modeling Data"

**Response**

"2.3 data" has been replaced by "Modelling data" (Line 192).

(3) Lines 448: The sentence "The parametric prior distribution is calculated as" misses punctuation, should be "The parametric prior distribution is calculated as:"

**Response**

This punctuation has been modified in the revised version (Line 514).

(4) Lines 475: Usually the results and discussion of scientific papers are written separately, it is recommended that this manuscript is also separated.

**Response**

Separate Results and Discussion sections have been written (Lines 541 and 662).

**General comments**

Review for -"A physically-based distributed karst hydrological model (QMG model-V1.0) for flood simulations" by Ji Li et.

It is well known that numerical simulation of karst groundwater is very difficult. Because of the complex karst basin subsurface conditions and spatially anisotropic karst water-bearing medias, it is a great challenge to quantitatively describe the transport and transformation patterns of karst groundwater. This study proposed a distributed hydrological model- QMG model for flood simulation and prediction in karst regions. This QMG model is a karst groundwater numerical model with potential application value, and the reasonable flood simulation results proved the accuracy and applicability of the model. The main innovation of this study is that a new karst model (QMG model-V1.0) is proposed and good flood simulation results are obtained. In addition, I am interested in the fact that the code of this model is very simple and easy to operate. Through my own modeling test, I used the modeling data in the case provided by the author to complete the construction and trial operation of the QMG model, and obtained satisfactory flood simulation results, which proved the effectiveness of the model. However, there are some language and writing problems that hinder readability and fluency of this paper, and I think the current version needs some minor revisions before it can be published in the GMD journal.

**Response**

Thank you for your praise regarding the potential application of the QMG model presented in this study. The point-by-point responses to your comments are as follows.

**Specific comments**

**1) Language and writing problems**

There are some English language problems in the current manuscript, including wrong words, grammar and inappropriate technical terms. For instance, "forecasting" is used many times in the paper, and I think the potential use of this new QMG model is better written as a "prediction"

based on the simulation results of this study. It is suggested that the authors find a hydrologist who is a native English speaker to help revise the language and writing problems of the whole paper.

**Response**

We have carefully revised the language issues throughout the text and enlisted the help of a professional editing service (American Journal Experts). Some technical terms such as "forecasting" have been replaced by "prediction".

**2) Abstract**

It is suggested to add some technical indexes to evaluate the performance of the new model in the abstract, so as to better reflect the effect of the QMG model in karst flood simulations.

**Response**

Some technical indices have been added to the abstract to evaluate the model performance (Lines 31).

**3) Keywords**

"Simulation and forecasting of karst floods" should be replaced by "Simulation and prediction of karst floods".

**Response**

**This phrase has been modified accordingly (Line 36).**

**4) Introduction**

I read through the whole text and found that the latest literature cited by the authors is only from 2021. In the last two years 2020-2022 many hydrologists have also published some important literature on theoretical development and application of hydrological models in karst areas, and I suggest the authors to add several representative ones to the Introduction and the References lists. For instance,

Masciopinto, C., Passarella, G., Caputo, M. C., Masciale, R, & Carlo, L. D.. (2021). Hydrogeological models of water flow and pollutant transport in karstic and fractured reservoirs. Water Resources Research, 57. Zhang, H. (2021). Characterization of a multi-layer karst aquifer through analysis of tidal fluctuation. Journal of Hydrology, 601, 126677.

Gautama, R.S., Notosiswoyo, S., Zen, M. T., & Kusumayudha, S. B. (2021). Mathematical model of fractal conduits flow mechanics in the gunungsewu karst area, yogyakarta special region, indonesia. International Journal of Hydrology Science and Technology, 1(1), 1.

Y Chang, Hartmann, A., Liu, L., Jiang, G., & Wu, J. (2021). Identifying more realistic model structures by electrical conductivity observations of the karst spring. Water Resources Research.

Response

Some relevant and important new literature has been added to the revised paper, for instance, Gautama et. (2021) in Line 43; Masciopinto et. (2021) in Line 104; Chang et. (2021) in Line 75; Zhang (2021) in Line 62; Jamal and Awotunde (2022) in Line 65; and Guila et. (2022) in Line 75. These studies have been added to the References list.

5) Study area and data

"2.3 data" should be replaced by "2.3 Modeling Data". Considering that the audience of the article may not be hydrology professionals, it is recommended that the author try to put himself in the position of a lay reader when writing, so as to ensure that the written paper can be easily read and understood.

**Response**

"2.3 data" has been replaced by "2.3 Modelling Data" in Line 192.

6) Methodology

The parentheses in formula 1 are not formatted properly and need to be rewritten.

"3.1.2 Runoff generation" should be replaced by "3.1.2 Runoff generation algorithms".

"3.1.3 Channel routing and confluence" should be replaced by "3.1.3 Confluence algorithms".

**Response**

"3.1.2 Runoff generation" has been replaced by "3.1.2 Runoff generation algorithms" (Line 290).

"3.1.3 Channel routing and confluence" has been replaced by "3.1.3 Confluence algorithms" (Line 330).

7) Results and discussion

It is recommended to write these two parts separately, means 4 Results and 5 Discussion.

**Response**

Separate Results and Discussion sections have been written.